



# Regional seesaw between North Atlantic and Nordic Seas during the last glacial abrupt climate events

Mélanie Wary[1], Frédérique Eynaud[1], Didier Swingedouw[1], Valérie Masson-Delmotte[2], Jens Matthiessen[3], Catherine Kissel[2], Jena Zumaque[1,4], Linda Rossignol[1], Jean Jouzel[2]

[1]UMR 5805, EPOC (Environnements et Paléoenvironnements Océaniques et Continentaux), CNRS-EPHE-Université de Bordeaux, 33615 Pessac, France.
[2]UMR8212, LSCE (Laboratoire des Sciences du Climat et de l'Environnement)/IPSL (Institut Pierre Simon Laplace), CEA/CNRS-INSU/UVSQ, 91191 Gif-sur-Yvette CEDEX, France.
[3]AWI (Alfred Wegener Institute), Helmholtz Centre for Polar and Marine Research, 27568 Bremerhaven, Germany.
[4]Now at GEOTOP, UQAM, Montréal, Québec H3C 3P8, Canada.

*Correspondance to:* Mélanie Wary (melanie.wary@u-bordeaux.fr)

**Abstract.** Dansgaard-Oeschger oscillations constitute one of the most enigmatic features of the last glacial cycle. Their cold atmospheric phases have been commonly associated with cold sea-surface temperatures and expansion of sea ice in the North Atlantic and adjacent seas. Here, based on dinocyst analyses from the 48-30 ka BP interval of four sediment cores from the northern Northeast Atlantic and southern Norwegian Sea, we provide direct and quantitative evidence of a regional paradoxical seesaw pattern: cold Greenland and North Atlantic phases coincide with warmer sea-surface conditions and shorter seasonal sea-ice cover durations in the Norwegian Sea as compared to warm phases. Combined with additional paleorecords and multi-model hosing simulations, our results suggest that during cold Greenland phases, reduced Atlantic meridional overturning circulation and cold North Atlantic sea-surface conditions were accompanied by the subsurface propagation of warm Atlantic waters that re-emerged in the Nordic Seas and provided moisture towards Greenland summit.

## 1 Introduction

The last glacial cycle has been punctuated by abrupt climatic variations strongly imprinted in Greenland ice core records where they translate into millennial oscillations between cold (Greenland stadial, GS) and warm (Greenland interstadial, GI) atmospheric phases (e.g., North Greenland Ice Core Project members, 2004). They are tightly linked to pan-North Atlantic ice-sheet dynamic that manifests itself by cyclic iceberg releases concomitant with GS (Bond and Lotti, 1995). These variations are thought to be linked to changes in the North Atlantic meridional overturning circulation, potentially in response to iceberg-derived freshwater injections in the North Atlantic (Kageyama et al., 2010). A few paleoclimatic studies (Dokken and Jansen, 1999; Rasmussen and Thomsen, 2004; Dokken et al., 2013) and sensitivity tests performed with atmospheric models (Li et al., 2010) have also suggested that the expansion of sea ice in the Nordic Seas during GS could be a key amplifier, explaining the large 5-16 °C magnitude of Greenland cooling (Kindler et al., 2014). However, cold sea-surface temperatures (SST) and expansion of sea ice during GS were mainly inferred from indirect marine proxy records, such as significant increases in ice-rafted debris concentration or variations in the relative abundance





and oxygen isotopic content of the polar planktonic foraminifera *Neogloboquadrina pachyderma* sinistral coiling (NPS) (Bond and Lotti, 1995; Dokken and Jansen, 1999; Rasmussen and Thomsen, 2004; Dokken et al., 2013) whose preferential depth habitat lies from a few tens of meters to around 250 meters water depth in the Nordic

Seas (Simstich et al., 2003). Furthermore, the few direct but qualitative sea-ice reconstructions based on lipid biomarker analyses (Müller and Stein, 2014; Hoff et al., 2016) yielded contrasting results. Here, we provide direct and quantitative reconstructions of variations of sea-surface conditions from a compilation of three Norwegian Sea cores and one northern Northeast Atlantic core strategically positioned across the Faeroe-Iceland Ridge to track rapid hydrographic changes (Dokken and Jansen, 1999; Eynaud et al., 2002; Rasmussen and

Thomsen, 2004; Dokken et al., 2013) (Fig. 1A and Table S1). We focus on Marine Isotopic Stage 3 (MIS 3, 30-48 ka cal BP), when millennial variability is strongly imprinted, and accurate chronologies can be established (Austin and Hibbert, 2012). In parallel to these reconstructions, we also use subsurface hydrographical data, freshwater hosing simulations and ice core-derived atmospheric data to assess the ocean-cryosphere-atmosphere interactions associated with this abrupt climate variability.

## 2        Methods

### 2.1        Stratigraphy

For the four studied cores, new age models have been established on the basis of radiocarbon AMS $^{14}$C dates coupled to additional tie-points obtained by correlating their magnetic susceptibility records with the NGRIP δ$^{18}$O signal (North Greenland Ice Core Project members, 2004) (GICC05 time scale; Svensson et al., 2008). This

approach is in line with the current consensus that, in this region, increases (respectively decreases) in magnetite content (here, magnetic susceptibility reflecting deep sea currents strength; Kissel et al., 1999) are synchronous with the onset of GI (respectively onset of GS; Kissel et al., 1999; Austin and Hibbert, 2012). Cores MD95-2009, MD95-2010 and MD99-2281 also benefit from additional climate-independent age control points supporting these new age models. A more detailed discussion on the age models can be found in the Supporting

Information (Section S1, Fig. S1, and Table S2; Martinson et al., 1987 ; Manthé, 1998; Laj et al., 2004 ; Rasmussen et al., 2006 ; Zumaque et al., 2012 ; Caulle et al., 2013; Reimer et al., 2013; Wolff et al., 2010; Wary et al., 2016).

### 2.2        Sea-surface conditions

Sea-surface conditions are estimated from a transfer function applied to dinocyst – or dinoflagellate cyst –

assemblages using the modern analogue technique (de Vernal and Rochon, 2011) (see Supporting Information Section S2 for further details on the methodology; Rochon et al., 1999; Head et al., 2001; Radi et al., 2013, Guiot and de Vernal, 2007, 2011a, 2011b). As dinoflagellates are mostly restricted to the uppermost 50 meters water depth (Sarjeant, 1974), they are assumed to directly reflect sea-surface conditions (see Supporting Information Section S6 for further details). This statistical approach provides direct and quantitative

reconstructions for mean summer and mean winter SST (with root mean square errors of prediction – RMSEP – of 1.5 °C and 1.05 °C respectively), mean summer and mean winter sea-surface salinities (SSS; respective RMSEP of 2.4 and 2.3 psu), and mean annual sea-ice cover (SIC) duration (RMSEP of 1.2 month/year).



### 2.3 Model simulations

We compare our reconstructions with freshwater hosing experiments conducted using five state-of-the-art
climate models (Swingedouw et al., 2013). Four of them are coupled ocean-atmosphere models (HadCM3,
IPSLCM5A, MPI-ESM, EC-Earth) and one is ocean-only model (ORCA05) (see Supporting Information
Section S3 and Table S3; Gordon et al., 2000; Biastoch et al., 2008; Sterl et al., 2012; Dufresne et al., 2013;
Jungclaus et al., 2013). Two types of simulations are considered: (i) the transient control simulations,
corresponding to historical simulations without any additional freshwater input, and (ii) the hosing simulations,
corresponding to historical simulations with an additional freshwater input of 0.1 Sv released on all the coastal
grid points around Greenland with a homogenous rate during 40 years (over the historical era 1965–2004, except
for HadCM3 and MPI-ESM for which the experiments were performed over the periods 1960–1999 and 1880–
1919, respectively). Several variables have been analyzed: oceanic temperatures (Fig. 1B and 1D), surface (2 m)
atmospheric temperatures (Fig. 1C), and barotropic stream function (Fig. S6). Anomalies were calculated as the
differences between hosing and control experiments averaged over the 4th decade.

Earlier studies have shown that the response (spatial pattern, amplitudes, …) to freshwater discharges in the
North Atlantic depends on several factors including climatic boundary conditions (Swingedouw et al., 2009;
Kageyama et al., 2010). Differences of sensitivity to freshwater perturbations in Last Glacial Maximum (LGM)
conditions compared to interglacial conditions have been mainly ascribed to differences in ice-sheet and sea-ice
configurations. As millennial climatic variability is strongest during MIS 3, it would have been optimal to
compare our MIS 3 data to simulations run under MIS 3 conditions rather than pre-industrial ones. However,
MIS 3 boundary conditions, and especially cryospheric conditions, are poorly constrained and set at an
intermediate level between LGM and present-day boundary conditions (Van Meerbeeck et al., 2009).
Nevertheless, it will be worth comparing our reconstructions with MIS 3 simulations conducted using the same
state-of-the-art multi-model approach with standardized volume and duration of freshwater flux as soon as such
simulations will be available.

### 2.4 Complementary data

To complement our view of the system, we also compare our sea-surface hydrographical reconstructions with (i)
the relative abundance of the mesopelagic polar planktonic foraminifera NPS (Eynaud et al., 2002; Zumaque et
al., 2012; Wary, 2015) as tracer of cold subsurface conditions (see Supporting Information Sections S5 and S6
for further details; Carstens and Wefer, 1992; Bauch et al., 1997; Carstens et al., 1997; Hillaire-Marcel and
Bilodeau, 2000; Volkmann and Mensch, 2001; Simstich et al., 2003; Hillaire-Marcel et al., 2004; Kretschmer et
al., 2016), and (ii) Greenland ice core deuterium excess data as indicator of Greenland moisture origin (Masson-
Delmotte et al., 2005).

### 3 Results and Discussion

Our sea-surface reconstructions reveal contrasted responses of the southeastern Nordic Seas compared to the
northeastern Atlantic (Fig. 2, Tables 1 and 2). The Atlantic core MD99-2281 exhibits lower SST during GS
compared to GI (Fig. 2), and a very short SIC duration throughout MIS3. Surprisingly, the three Norwegian Sea





cores record higher SST and shorter SIC durations during the cold North Atlantic GS, and lower SST and longer

SIC durations during the warm North Atlantic GI. This atypical pattern is robustly observed in all the three Norwegian Sea sequences, despite distinct physiographical contexts, and strongly expressed in the 63°N cores. At this latitude, SST is systematically anti-correlated against Greenland and North Atlantic temperatures (correlation coefficients of -0.45 between NGRIP $\delta^{18}$O and MD99-2285 winter SST, -0.43 between NGRIP $\delta^{18}$O and MD99-2285 winter SST, and -0.31 between Atlantic site MD99-2281 winter SST and both MD99-2285 and

MD95-2009 winter SST), and shows large positive mean annual anomalies in GS compared to GI from +1.7 °C (MD95-2009) to +3.7 °C (MD99-2285) (see Supporting Information Section S4 for details on the calculation of anomalies; Wolff et al., 2010). Despite lower resolution and sensitivity (Eynaud et al., 2012), SST records from MD95-2010 also denote a positive GS mean annual SST anomaly (+0.9 °C), and cooling during GI is further supported by increases in the relative percentage of the polar, sea-ice linked dinocyst *Islandinium minutum* (%

I.MIN; Supporting Information Section S2 and Figs. S2 and S3; Rochon et al., 1999; Radi et al., 2013). Previous paleoclimatic studies (e.g. de Vernal et al., 2006) evidenced a similar regional SST seesaw pattern during the LGM, suggesting that such a situation might represent a regular feature for glacial periods.

In order to investigate the mechanisms involved in this regional seesaw, we analyze the multi-model freshwater hosing simulations from Swingedouw et al. (2013). The five-member ensemble mean of the differences between

hosing and control experiments shows large surface warming in the Nordic Seas while the rest of the North Atlantic surface is strongly cooled in response to freshwater input around Greenland (Fig. 1B). This regional seesaw pattern is robust in the five individual simulations and consistent with concomitant atmospheric cooling above Greenland (Fig. 1C). While the simulated multi-model mean surface warming is weaker than the paleodata-derived one, some individual simulations produce SST increase of up to 4.2 °C in the Nordic Seas

(Swingedouw et al., 2013). The multi-model simulations also depict significant sea-ice retreat in the Nordic Seas and sea-ice expansion in the Atlantic sector and Labrador Sea (see Fig. 10 in Swingedouw et al., 2013).

An earlier modelling study (Kleinen et al., 2009) also depicted surface warming of the Nordic Seas in response to a freshwater perturbation, independently from the location of the freshwater input. It was attributed to the subsurface propagation of warm Atlantic water masses beneath the cold North Atlantic meltwater lid (resulting

from the freshwater input) up to the Nordic Seas where they re-emerge and mix with ambient waters. Our model simulations indeed show a positive subsurface heat anomaly south of the Greenland-Scotland sill, located below the North Atlantic freshwater lid (Fig. 1D). This freshwater lid has two important consequences: (i) it prevents oceanic vertical mixing which normally transfers winter surface cooling (due to atmospheric heat fluxes) into subsurface, and (ii) it induces hydrographical reorganizations where subpolar gyre transport decreases but water-

mass transport from the subtropics into the Nordic Seas increases, especially along the eastern North Atlantic boundary (see Hátún et al., 2005; Kleinen et al., 2009) and Fig. S6). Although simulated here under present-day background conditions, this physical process may have occurred during stadials in response to meltwater release and provides an explanation for the regional seesaw SST and SIC pattern.

We now consider subsurface information from our records to complement this mechanism (Fig. 2). Consistent

with earlier paleoceanographic studies within the Nordic Seas (Rasmussen and Thomsen, 2004) and the North Atlantic (Bond and Lotti, 1995; Rasmussen and Thomsen, 2004; Eynaud et al., 2009; Jonkers et al., 2010), all our cores reveal the occurrence of colder planktonic foraminiferal assemblages during GS, characterized here by



nearly 100% of the mesopelagic taxon NPS. This testifies to the presence of cold polar waters (Eynaud et al., 2009) below a few tens of meters of water depth.

Altogether, this implies the following oceanographic situation during GS: a reduced Atlantic meridional overturning circulation due to large meltwater fluxes (related to and/or sustained by iceberg releases), a southward migration of polar waters, a colder and fresher North Atlantic surface, and a small northward subsurface flow of warm Atlantic waters, propagating below the North Atlantic meltwater lid (and below NPS depth habitat) before remerging at the surface of the Nordic Seas, above colder polar waters (Fig. 3).

The upper part of the water column (topmost tens of meters) consists of a layer characterized by fairly high temperatures, notably during summer (Table 1), due to increased heat transport associated with Atlantic waters without heat loss. Dinocyst-derived sea-surface salinities (Table S4) also depict relatively low values, around 31.7 psu over the entire study area. These low salinities are probably due to (i) surface meltwater produced by iceberg releases within the Nordic Seas, evidenced by ice-rafted peaks during GS (Elliot et al., 2001), and (ii) the

seasonal melting of (reduced) sea ice. At the base of this warm and low saline layer, the nearly 100% NPS indicates colder (at least during summer) and probably slightly saltier waters than in the upper layer (Tolderlund and Bé, 1971).

Using indirect proxies, earlier studies (Rasmussen and Thomsen, 2004; Dokken et al., 2013) had suggested the existence of a strong pycnocline separating cold and fresh surface waters from warm and salty Atlantic

subsurface waters during GS. Our direct reconstructions depict a more complex temperature-depth pattern but also imply a pycnocline. This stratification of the upper water column results in strong sea-surface seasonality contrasts as depicted by dinocysts during GS (Supporting Information Section S2 and Fig. S4; Locarnini et al., 2010). They are explained by the relatively low thermal inertia of the low salinity surface waters, and the limited winter sea-ice extent. Sea-ice cover duration is less than 3.5 months/year at the study sites. Reduced sea-ice

formation during GS compared to GI possibly relates to the heat transport by the Atlantic waters, in an orbital context during MIS3 with high summer insolation at 65°N (Berger and Loutre, 1991).

During GI (Fig. 3), coherent sea-surface and subsurface patterns are reconstructed in the four sediment cores, reflecting the disappearance or deepening of the pycnocline. The Norwegian Sea is then characterized by lower SST, reduced seasonal SST contrasts, and 100% NPS, reflecting a thick homogenous mixed layer consisting of

cold polar waters, as well as longer sea-ice cover durations. In the Atlantic sector, core MD99-2281 exhibits less than 50% NPS, higher SST and reduced seasonal SST contrasts, indicating a thick and weakly stratified mixed layer where polar waters and Atlantic waters mix.

Our new paradigm is thus consistent with a scenario of subsurface and intermediate-depth warming during GS in the North Atlantic (Jonkers et al., 2010; Marcott et al., 2011) and in the Nordic Seas (Rasmussen and Thomsen,

2004; Marcott et al., 2011; Dokken et al., 2013; Ezat et al., 2014), where reconstructed subsurface and bottom temperatures are quite lower than our reconstructed summer SST. Such subsurface warming might be due to the insulation by the North Atlantic meltwater lid and downward diffusion of heat in the Nordic Seas.

It is not incompatible with the "brine hypothesis" (Dokken and Jansen, 1999; Dokken et al., 2013) formulated to explain the isotopically light $\delta^{18}O$ values measured on NPS during GS within cores from the southern Nordic

Seas, including core MD95-2010 (Dokken and Jansen, 1999), if we take into account changes of upper



stratification during GS/GI and seasonality of NPS production period in the Nordic Seas (Simstich et al., 2003). During GS (strong stratification), NPS $\delta^{18}O$ may reflect reduced winter shelf brine production – stored within the subsurface layer inhabited by NPS – rather than the seasonal melting, trapped in surface. During GI (weak stratification), NPS $\delta^{18}O$ may then only reflect the large summer melting of sea ice which produces isotopically heavier waters (Hillaire-Marcel and de Vernal, 2008). It is worth noting that the isotopically light brine extrusion is produced during winter, when NPS is nearly absent, and is expected to form bottom waters through convective processes without stagnating at the base of the mixed layer.

The reconstructed SST pattern has implications for atmospheric circulation, moisture sources, and interpretation of Greenland ice core water stable isotope records, especially deuterium excess data (Masson-Delmotte et al., 2005) (Fig. 2). Recent monitoring data have revealed that (i) deuterium excess is low for subtropical Atlantic vapor and high for vapor formed at Arctic sea ice margin where high kinetic fractionation occurs due to low relative humidity, and (ii) this vapor deuterium excess is preserved during transportation towards Greenland (Jouzel et al., 2013; Bonne et al., 2015). Higher deuterium excess recorded during GS (Masson-Delmotte et al., 2005) may reflect enhanced contribution of moisture from the Nordic Seas towards Greenland, when the Norwegian Sea appears relatively warm and surrounded by sea-ice covered areas (providing low humidity air masses), while the North Atlantic surface is cold and marked by large sea-ice expansion (Hillaire-Marcel and de Vernal, 2008).

## 4     Conclusion

Our description of regional patterns and oceanographic processes occurring during MIS3 within the North Atlantic and the Nordic Seas is thus consistent with all existing paleoclimate information and with climate simulations in response to freshwater forcing. During GS, we evidence large surface warming within the Nordic Seas, in response to high-latitude freshwater release and subsequent regional reorganizations. Such warming might have enhanced iceberg releases from the bordering ice-sheets, and might have therefore constituted a positive feedback for freshwater release. The origin of the freshwater-forcing input is still enigmatic, and may be related to, or precede (Barker et al., 2015), massive iceberg calving episodes. Our findings thus highlight an original case study for climate – ice sheet interactions, and calls for additional numerical simulations focused on ocean – sea ice – atmosphere interactions during MIS 3 millennial climatic events. As a first step, evidencing such a warming of the Nordic Seas in response to a standardized freshwater release in the subpolar gyre in an ensemble of state-of-the-art climate models under MIS3 conditions will be a prerequisite.



**Acknowledgements**

We thank M.-H. Castéra for technical assistance, D. Roche, F. Ziemen, U. Mikolajewicz, M. F. Sánchez Goñi, M. Kageyama, M. Elliot, A. Penaud and F. Marret-Davies for discussions, and S. Manthé for MD95-2009 and MD95-2010 planktonic foraminiferal counts. Analyzes performed on MD99-2285 were supported by French INSU (Institut National des Sciences de l'Univers) programme LEFE (Les enveloppes fluides et

l'environnement) within the frame of the EVE (Evolution et variabilité du climat à l´échelle globale) 2009-2011 project "RISCC: Rôle des Ice-Shelves dans le Changement Climatique" and of the IMAGO (Interactions multiples dans l'atmosphère, la glace et l'océan) 2013 project "ICE-BIO-RAM : Impact des Changements Environnementaux sur la BIOdiversité marine lors des Réchauffements Abrupts du cliMat", this latter being also supported by the INTERRVIE (Interactions Terre/Vie) - TS (Terre solide) INSU programme. We also

acknowledge financial supports and facilities of the ARTEMIS 14C AMS French INSU project. The research leading to these results also benefited from fundings from the European Union's Seventh Framework programme (FP7/2007-2013) under grant agreement no 243908, "Past4Future. Climate change - Learning from the past climate" and from Agence Nationale de la Recherche (ANR) "Greenland" project (grant ANR-10-CEPL-0008).

Data used in this study are available upon request to Mélanie Wary (melanie.wary@u-bordeaux.fr) and

Frédérique Eynaud (frederique.eynaud@u-bordeaux.fr).

The authors declare that they have no conflict of interest.





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



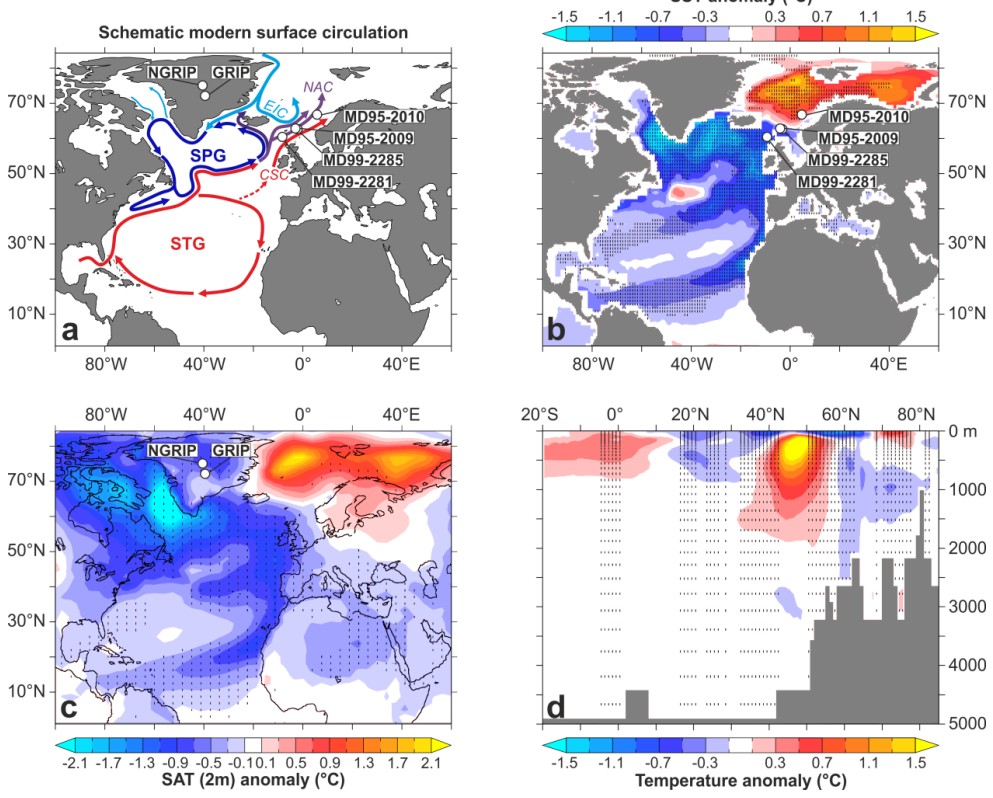

**Figure 1: Hydrographical context and five-member ensemble mean of temperature anomalies between hosing and control experiments. (a) Schematic surface current pattern (STG, subtropical gyre; SPG, subpolar gyre; CSC, Continental Slope Current; NAC, North Atlantic Current; EIC, East Icelandic Current). (b, c) Five-member ensemble mean of SST (b) and surface atmospheric temperature (c) anomalies (°C). (d) Latitude-depth section of the five-member ensemble mean of oceanic temperature anomalies (°C, zonal average over Atlantic ocean). Also shown are the locations of the studied marine cores (MD95-2010, MD95-2009, MD99-2285, MD99-2281) and Greenland ice cores (NGRIP, GRIP). Black dashes indicate grid points where all models converge on the anomaly sign.**



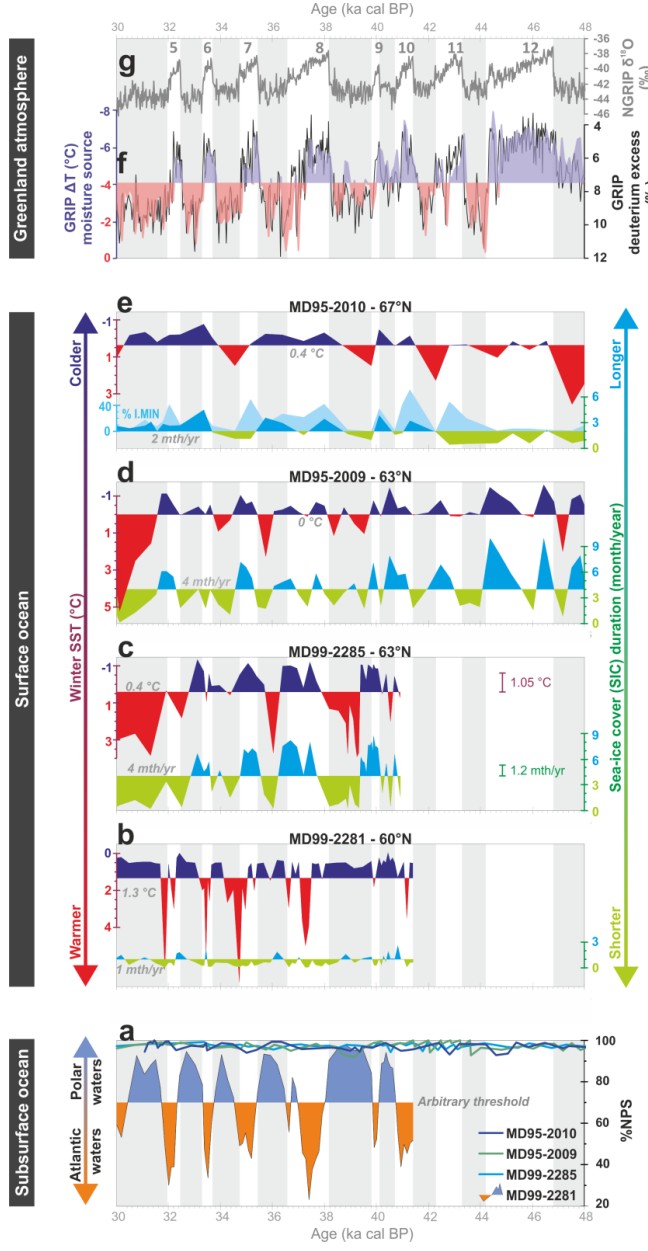

**Figure 2: Proxy records. (a) %NPS (shading in MD99-2281 at an arbitrary threshold to better illustrate changes). (b to e) SST and SIC records, shaded relatively to the mean value (indicated in gray) of the parameter over the studied period. Error bars are shown in panel (c). %I.MIN of MD95-2010 is also shown. (f) GRIP deuterium excess record and associated reconstructed source temperature anomaly (compared to modern value) of the evaporative source region for Greenland precipitation, assuming no change in humidity (Masson-Delmotte et al., 2005) (shaded relatively to its mean value over the studied period). (g) NGRIP δ¹⁸O (GICC05 age scale; North Greenland Ice Core Project members, 2004; Svensson et al., 2008). Gray bands highlight stadial periods.**


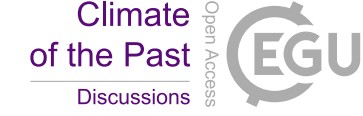

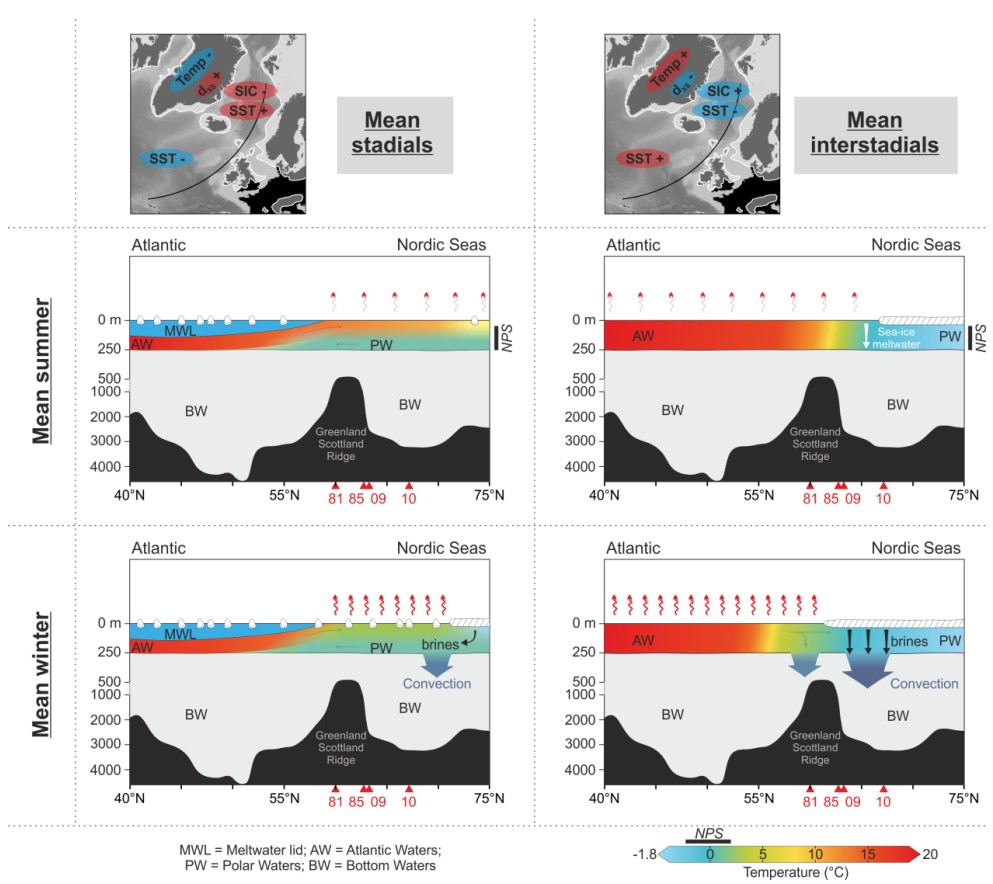


**Figure 3: Conceptual hydrographical scheme. The diagrams depict the mean conditions in the subboreal Atlantic during stadials (left) /interstadials (right), and summer (middle panels) /winter (lower panels). Section location is indicated on the top maps. Bathymetry is from GEBCO (www.gebco.net), and has been simplified for sections. Ice-sheet extent on maps corresponds to the Last Glacial Maximum extension (Ehlers and Gibbard, 2007). Colors indicate**

**temperature range, as indicated by the bottom scale. Potential depth range (Simstich et al., 2003) and optimal temperature range (Tolderlund and Bé, 1971) of NPS habitat, whose main production period occurs in summer in the Nordic Seas (Simstich et al., 2003), are also indicated.**





**Table 1: SST anomalies.**

| Core | Number of samples | | GS SST (°C) | | | GI SST (°C) | | | Mean annual SST anomalies (GS-GI; °C) |
|---|---|---|---|---|---|---|---|---|---|
| | GS | GI | mean winter | mean summer | mean annual | mean winter | mean summer | mean annual | |
| MD99-2281 | 23 | 39 | 0.9 | 14.6 | 7.8 | 1.5 | 14.4 | 8.0 | -0.2 |
| MD99-2285 | 26 | 22 | 0.9 | 10.9 | 5.9 | -0.6 | 4.9 | 2.2 | 3.7 |
| MD95-2009 | 12 | 17 | 0.3 | 11.0 | 5.6 | -0.4 | 8.3 | 4.0 | 1.7 |
| MD95-2010 | 6 | 9 | 0.6 | 13.4 | 7.0 | 0.2 | 12.4 | 6.1 | 0.9 |






Table 2: SIC duration anomalies.

| Core | Number of samples | | GS SIC (mth/yr) | GI SIC (mth/yr) | Mean annual SIC anomalies (GS-GI; mth/yr) |
|---|---|---|---|---|---|
| | GS | GIS | mean annual | mean annual | |
| MD99-2281 | 23 | 39 | 0.9 | 0.6 | 0.3 |
| MD99-2285 | 26 | 22 | 3.2 | 6.2 | -3.0 |
| MD95-2009 | 12 | 17 | 3.4 | 4.4 | -1.0 |
| MD95-2010 | 6 | 9 | 2.0 | 2.7 | -0.7 |