# Peer review of "Regional seesaw between North Atlantic and Nordic Seas during the last glacial abrupt climate events"

_Climate of the Past, 2017_

## Referee Comment (RC1) · Anonymous Referee #1 · 13 Mar 2017

This paper presents interesting reconstructions of sea-surface conditions in the North Atlantic and Nordic Seas for Marine Isotopic Stage 3, based on dinocyst assemblages and planktonic forams, as well as climate modelling. This is a very well written paper, well presented and argued, with little to fault. The only aspect I would like to have seen being discussed is the possible forcing of productivity on dinocyst assemblages, in particular the high abundance of I. minutum, which is recognized as a tracer of sea-ice cover, but also abundant in high nutrient environments (see Zonneveld et al 2013). Further studies by Heikkilä et al (2014, 2016) also suggest a more complex response of this species to sea-ice environments. Based on these ecological findings, how would it affect your interpretation?

Heikkilä, M., Pospelova, V., Forest, A., Stern, G.A., Fortier, L., Macdonald, R.W. Dinoflagellate cyst production over an annual cycle in seasonally ice-covered Hudson Bay (2016) Marine Micropaleontology, 125, pp. 1-24 Heikkilä, M., Pospelova, V., Hochheim, K.P., Kuzyk, Z.Z.A., Stern, G.A., Barber, D.G., Macdonald, R.W. Surface sediment dinoflagellate cysts from the Hudson Bay system and their relation to freshwater and nutrient cycling (2014)
* * *

---

## Author Comment (AC1) · 21 Mar 2017

The comment was uploaded in the form of a supplement:
http://www.clim-past-discuss.net/cp-2017-14/cp-2017-14-AC1-supplement.pdf

—————————————————

---

## Referee Comment (RC2) · Anonymous Referee #2 · 22 Mar 2017

Wary et al present an interesting compilation of (and new) sea surface temperature, salinity, sea ice cover reconstructions from the Norwegian Sea and northern North Atlantic based on dinocyst analyses for Marine Isotope Stage 3 as well as an ensemble of freshwater hosing experiments run under preindustrial boundary conditions. The paper is well written and data are very well presented and adds to the debate about the stadial/interstadial evolution of the Nordic Seas circulation during the last glacial and its role in the abrupt climate change. However, the paper needs moderate/major revisions before it could be accepted for publication.

1- First of all the authors need to elaborate on how summer SST up to 14 °C in the Norwegian Sea during stadials compare with other previous reconstructions. In this

regard, the following points need further discussion:

- The authors stated 'Furthermore, the few direct but qualitative sea-ice reconstructions based on lipid biomarker analyses (Müller and Stein, 2014; Hoff et al., 2016) yielded contrasting results'. Looking at Figure 4 in Hoff et al 2016, it does not seem that those two studies are at odd. In contrary, the sea ice cover records in Müller and Stein, 2014 and Hoff et al., 2016 seem for me to correlate well. I think it is critical to discuss why sea ice cover reconstructions in the southern Norwegian Sea in this study (dinocyst-based) and in Hoff et al., 2016 (lipid biomarker- based) significantly differ. I suggest you plot IP25, brassicasterol- and dinosterol concentration (not the PBIP25 and PDIP25 indexes) with your dinocyst-based data and see if you can reconcile between them or at least make the apparent disagreement between the two results clear, so future investigations may take it further.

- The authors may need to explain why the %subpolar planktic foraminifera (e.g., T. quinqueloba and G. bulloides) did not increase during stadials if summer SST was that high in the Norwegian Sea. I think the conditions at the average calcification depth of N. pachyderma may be best recorded in the isotopic and elemental composition of its shells, whereas the % N. pachyderma is also controlled by the abundance of other planktic species. For example, Mg/Ca in N. pachyderma shows different pattern from % N. pachyderma for Heinrich Stadial 1, also in the southern Norwegian Sea (Ezat et al., 2016).

- Notably, the reconstructed summer temperatures during glacial stadials in the southern Norwegian Sea in this study are similar to or even higher than modern temperatures. It is not plausible that we ignore an observation just because it does not fit with what we may expect. However, more discussion needed regards the temperature at the source of these water, were the stadial temperatures at lower latitudes higher than modern? In addition, the inflowing water may have had to mix with more cold polar water than in the modern case in its way to the Nordic seas.

2- It is important to clearly clarify in the methods (in section 2.2) what new data have been generated in this study and what have been used from previous studies. I think most of the dinocyst analyses from cores MD95-2009, MD95-2010 and MD99-2285 are already published, or? I may have just missed the referring to previous studies, so I hope this comment does annoy the authors if that is the case. Related to this, I would suggest removing the word 'Surprisingly' from the following sentence "Surprisingly, the three Norwegian Sea cores record higher SST and shorter SIC durations during the cold North Atlantic GS, and lower SST and longer SIC durations during the warm North Atlantic GI." This is not a surprise as previous dinocyst- based studies have already showed this stadial/interstadial SST pattern in the Norwegian Sea (e.g., Eynaud et al., 2002; Wary et al., 2016).

3- Line 139-144: Weakening of the subpolar gyre has been employed to explain relative warming in the eastern Nordic Seas under interglacial conditions for example during late Eemian (e.g., Born et al., 2011). However, a key difference here (in addition to many others), is the likely significant suppression of deep water formation in the Nordic Seas during MIS3 stadials, which has an impact of the inflowing surface water. So, adding more lines of discussions here is merited.

4- Lines 193–203: I think the discussion that enhanced contribution of moisture from the Norwegian Sea towards Greenland (inferred from SST reconstructions) may played a role in the increase in the deuterium excess recorded in Greenland ice cores during stadials......has discussed in Wary et al. 2016. If so, please summarize and add Wary et al., 2016 as a reference.

5- Minor issues:

- Please make sure that Müller and Stein, 2014 is included in the reference list.

- Supporting Information (Line 8): the authors may consider the use of shallow subsurface reservoir age estimates from the northern North Atlantic (e.g., Stern and Lisiecki, 2013; Thornalley et al., 2011) and from the Norwegian Sea (Ezat et al., 2016; Thornalley et al., 2015) to correct for past changes in reservoir ages.

References

Born, A., et al. (2011), Late Eemian warming in the Nordic Seas as seen in proxy data and climate models, Paleoceanography, 26, PA2207.

Eynaud, F., et al. (2002), Norwegian sea-surface palaeoenvironments of marine oxygen-isotope stage 3: the paradoxical response of dinoflagellate cysts. J. Quat. Sci. 17.

Ezat, M. M., et al. (2016), Reconstruction of hydrographic changes in the southern Norwegian Sea during the past 135 kyr and the impact of different foraminiferal Mg/Ca cleaning protocols, Geochem. Geophys. Geosyst., 17, 3420–3436.

Ezat, M. M., et al. (2017), Ventilation history of Nordic Seas overflows during the last (de) glacial period revealed by species‐specific benthic foraminiferal 14C dates, Paleoceanography 32, 172-181.

Hoff, U., et al. (2016), Sea ice and millennial-scale climate variability in the Nordic seas 90 ka to present, Nat. Commun., 7, 12247.

Müller, J. and Stein, R. (2014), High-resolution record of late glacial and deglacial sea ice changes in Fram Strait corroborates ice-ocean interactions during abrupt climate shifts. Earth Planet. Sci. Lett. 403, 446–455.

Stern, J. V., and L. E. Lisiecki (2013), North Atlantic circulation and reservoir age changes over the past 41,000 Źyears, Geophysical Research Letters, 40, 3693–3697.

Thornalley, D. J. R., et al. (2011), The Deglacial Evolution of North Atlantic Deep Convection, Science, 331, 202–205.

Thornalley, D. J. R., et al. (2015), A warm and poorly ventilated deep Arctic Mediterranean during the last glacial period, Science, 349, 706–710.‐

Wary, M., et al. (2016), Norwegian Sea warm pulses during Dansgaard-Oeschger stadials: Zooming in on these anomalies over the 35–41 ka cal BP interval and their impacts on proximal European ice-sheet dynamics, Quat. Sci. Rev., 151, 255–272.
* * *

---

## Author Comment (AC2) · 14 Apr 2017

RC2: Wary et al present an interesting compilation of (and new) sea surface temperature, salinity, sea ice cover reconstructions from the Norwegian Sea and northern North Atlantic based on dinocyst analyses for Marine Isotope Stage 3 as well as an ensemble of freshwater hosing experiments run under preindustrial boundary conditions. The paper is well written and data are very well presented and adds to the debate about the stadial/interstadial evolution of the Nordic Seas circulation during the last glacial and its role in the abrupt climate change. However, the paper needs moderate/major revisions before it could be accepted for publication.

Reply: We want to thank Anonymous Referee #2 for his/her careful and constructive review of our paper. We will take into account all his/her precious advice for the revision of the manuscript. Below are our replies to his/her comments.

1- First of all the authors need to elaborate on how summer SST up to 14°C in the Norwegian Sea during stadials compare with other previous reconstructions. In this regard, the following points need further discussion:

- The authors stated 'Furthermore, the few direct but qualitative sea-ice reconstructions based on lipid biomarker analyses (Müller and Stein, 2014; Hoff et al., 2016) yielded contrasting results'. Looking at Figure 4 in Hoff et al 2016, it does not seem that those two studies are at odd. In contrary, the sea ice cover records in Müller and Stein, 2014 and Hoff et al., 2016 seem for me to correlate well.

We totally agree, this figure (enclosed below) clearly shows the same results, i.e. low PIP25 values during Heinrich event 1 (and likely Heinrich event 2 in-between GI 3 and 2 – "IS3" and "IS2" in this figure). However, the interpretations of authors significantly differ:

- Müller and Stein (2014) interpreted the sharp decrease down to very low PIP25 values at the beginning of HE1 as the "**sudden break-up of the ice cover and the concentrated release of high amounts of sea ice and icebergs trapped therein**" (page 453, Section 6).

- In Hoff et al. (2016), as Heinrich event 1 is "characterized by IP25 and brassicasterol (as well as dinosterol) values of close to or zero", "PBIP25 and PDIP25 values of 1 were assumed" (page 4, legend Figure 4) and interpreted by the authors as "**a very prominent period of perennial or near-perennial sea ice cover**" (page 6).

The use of the word "results" was for sure inappropriate, and we will replace it by "interpretations".

[Figure]

**Figure 4 | Sea ice proxies for the time interval between 30 and 8 kyr ago.** Comparison of $P_BIP_{25}$ and $P_DIP_{25}$ signals of sediment core JM11-FI-19PC (GICC05-timescale) and sediment core MSM5/5-712-2 (calibrated age scale) from the Svalbard margin published by Müller and Stein[17]. For H-event 1 (blue bar), characterized by $IP_{25}$ and brassicasterol (as well as dinosterol) values of close to or zero (that is, 'zero divided by zero'), per definition maximum $P_BIP_{25}$ and $P_DIP_{25}$ values of 1 were assumed (for background see refs 15,26). For core locations, see Fig. 1. BA, Bølling and Allerød interstadial; H, H-event; HS, Heinrich stadial; LGM, Last Glacial maximum; YD, Younger Dryas.

*Figure 4 from Hoff et al. (2016)*

I think it is critical to discuss why sea ice cover reconstructions in the southern Norwegian Sea in this study (dinocyst-based) and in Hoff et al., 2016 (lipid biomarker- based) significantly differ. I suggest you plot IP25, brassicasterol- and dinosterol concentration (not the PBIP25 and PDIP25 indexes) with your dinocyst-based data and see if you can reconcile between them or at least make the apparent disagreement between the two results clear, so future investigations may take it further.

We also think it is critical to understand why our dinocyst-based sea-ice reconstructions and Hoff et al. (2016) biomarker-based sea-ice reconstructions significantly differ. But to do that properly, we think it is first needed to clarify why the same biomarker-derived signals lead to opposite interpretations (Hoff et al., 2016 *versus* Müller and Stein, 2014), and why the same biomarker proxy (IP25 abundances) can lead to opposite results in the same studied area (Hoff et al., 2016 on core JM11-FI-19PC *versus* Sicre, unpublished data on core MD99-2285, see figure p.194 of Wary, 2015: www.theses.fr/2015BORD0316/document). Unfortunately, we are not allowed to provide you with the plot of these MD99-2285 IP25 data along with Hoff et al. (2016)'s data, and we are truly sorry about that. But if you look at Figure 4b on p.194 of Wary (2015) and compare with Hoff et al. (2016)'s IP25 abundances (in µg/g of dry sed. for more coherency) you can see that both signals exhibit similar trends on the 39-36 ka BP section (i.e., higher IP25 abundances during GI8 and at the end of HS4, and lower ones during GS8), but rather different ones on the 41-39 ka BP interval (i.e. for core MD99-2285 (JM11-FI-19PC), lower (higher) IP25 abundances during GS10 (i.e. the "H4" IP25 peak in Hoff et al., 2016), higher (lower) IP25 abundances during GI9, and zero (low/moderate) IP25 abundances during HE4 interval as defined on the basis of our IRD data). In core MD99-2285, IP25 measurements were realized at very high temporal resolution (54 years on average) and they reveal a very noisy signal (but yet with clear trends), so the difference might maybe (or maybe not) come from this given the lower resolution in Hoff et al. (2016). But, whatever the reason(s) might be, we think that we are not the most appropriate co-author team to discuss that (no biomarker people among us), that our paper is not the most appropriate to discuss that (no biomarker data), and thus that it would be quite improper to discuss such discrepancies in our paper.

- The authors may need to explain why the %subpolar planktic foraminifera (e.g., T. quinqueloba and G. bulloides) did not increase during stadials if summer SST was that high in the Norwegian Sea. I think the conditions at the average calcification depth of N. pachyderma may be best recorded in the isotopic and elemental composition of its shells, whereas the %

N. pachyderma is also controlled by the abundance of other planktic species. For example, Mg/Ca in N. pachyderma shows different pattern from % N. pachyderma for Heinrich Stadial 1, also in the southern Norwegian Sea (Ezat et al., 2016).

We suggest that the % subpolar planktonic foraminifera did not increase during GS because: (i) in the surface layer, SSS were apparently too low (summer $SSS_{dino}$ between 30.9 and 31.6 on average) according to these species tolerances (e.g., Tolderlund and Bé 1971), and (ii) in the subsurface layer, temperatures were too low (below 5-6°C according to *N. pachyderma* s. optimal temperature range – e.g., Tolderlund and Bé 1971) according to these species tolerances, but these subsurface temperatures could have varied between 1°C and 5-6°C as shown in Ezat et al. (2016) and also suggested in Wary et al. (2016), in both cases for stadial intervals characterized by 100% NPS.

- Notably, the reconstructed summer temperatures during glacial stadials in the southern Norwegian Sea in this study are similar to or even higher than modern temperatures. It is not plausible that we ignore an observation just because it does not fit with what we may expect.

Modern SST values are indicated on Figure S4. We did not indicate them on Figure 2, because the SST range is too low to do so. But we can add these modern SST values on Figure S5 where annual, summer and winter SST are presented, and we can also mention it in the text as we already did in our previous papers.

Furthermore, comparable warm anomalies (as high as modern temperatures) were obtained (also on the basis of dinocyst reconstructions – but not exclusively – and of a multicore compilation) during the last glacial maximum (see de Vernal et al., 2006 for instance), suggesting an atypical and a much more complex functioning of the GIN seas at that time, quite different from the existing modern polar schemes we have in mind.

However, more discussion needed regards the temperature at the source of these water, were the stadial temperatures at lower latitudes higher than modern? In addition, the inflowing water may have had to mix with more cold polar water than in the modern case in its way to the Nordic seas.

According to our simulations, the initial freshwater flux induces a mid-latitude subsurface warming due to the meltwater lid, and the water mass transport is increased along the eastern North Atlantic boundary, potentially through the Continental Slope Current (flowing poleward along the European margin) as suggested in Wary et al. (2016). The source of this water is quite enigmatic, even if we consider that it is advected through this near-surface current whose origin is still debated (see Section 2 in Wary et al., 2016). Nonetheless, Peck et al. (2008) reported unexpectedly warm SST (sometimes warmer than modern ones) during Heinrich stadials at the Porcupine Seabight (~51°N) on the basis of Mg/Ca measurements on *G. bulloides* (see their Figure 4, enclosed below). They proposed several hypotheses to explain these too warm SST during Heinrich events, one of them being transient advection of subtropical surface waters through the Continental Slope Current.

[Figure]

**Figure 4.** Temperature estimates and seasonality of the upper water column at MD01-2461 over the last 55 ka. (a) GISPII $\delta^{18}O$ ice core record. (b) Summer SST inferred from Mg/Ca$_{G.\ bulloides}$ (red solid circles) following *Peck et al.* [2006]. Modern summer SST shown by red dashed line. Mean annual subsurface temperature inferred from Mg/Ca$_{N.\ pachyderma\ sinistral}$ (blue open circles) following *Peck et al.* [2006]. Modern mean annual water temperature at 150 m shown by blue dashed line. (c) June solstice solar insolation at 65°N (La90) [*Laskar*, 1990]. (d) $\Delta T$ (summer subsurface temperature (SST), °C). A 6th-order polynomial fit is shown in red. Modern $\Delta T$ and $\Delta T = 0$ shown with dashed lines.

*Figure 4 from Peck et al. (2008)*

2- It is important to clearly clarify in the methods (in section 2.2) what new data have been generated in this study and what have been used from previous studies. I think most of the dinocyst analyses from cores MD95-2009, MD95-2010 and MD99-2285 are already published, or? I may have just missed the referring to previous studies, so I hope this comment does annoy the authors if that is the case.

MD99-2281, MD95-2009 and MD95-2010 dinocyst assemblages are already published, as well as for core MD99-2285 on the 41-35 ka BP interval. For the present study, the published dinocyst counts of those cores and the new data of core MD99-2285 were used to generate new SST, SSS and SIC estimates thanks to the extended modern database used for the transfer function. We will move these information from the Supporting Information section to Section 2.2.

Related to this, I would suggest removing the word 'Surprisingly' from the following sentence "Surprisingly, the three Norwegian Sea cores record higher SST and shorter SIC

durations during the cold North Atlantic GS, and lower SST and longer SIC durations during the warm North Atlantic GI." This is not a surprise as previous dinocyst- based studies have already showed this stadial/interstadial SST pattern in the Norwegian Sea (e.g., Eynaud et al., 2002; Wary et al., 2016).

Our formulation is probably awkward, we do not mean that the results are surprising in the sense that they are new, but that the scheme described here at a regional scale is surprising because different from the accepted scheme. We will therefore reformulate our sentence.

3- Line 139-144: Weakening of the subpolar gyre has been employed to explain relative warming in the eastern Nordic Seas under interglacial conditions for example during late Eemian (e.g., Born et al., 2011). However, a key difference here (in addition to many others), is the likely significant suppression of deep water formation in the Nordic Seas during MIS3 stadials, which has an impact of the inflowing surface water. So, adding more lines of discussions here is merited.

In Born et al. (2011) deep water convection remains active in the Nordic Seas during the episode (119-115 ka BP) of weakening of the subpolar gyre circulation / increased heat transport in the Nordic Seas through the Norwegian Atlantic Current (NwAC, i.e. probable continuation of the Continental Slope Current). According to the authors, this is because even if "enhanced salt transport counteracts the general freshening […] locally but does not reverse it" and "despite increased heat transport by the NwAC, no absolute warming of the Nordic Seas is expected at 115 ka because of the counteracting large insolation forcing".

In our MIS3 case, it seems to be the same situation for the surface freshening since our reconstructed GI-GS SSS anomalies are quite small (0.0 to 0.4 psu). However, the scheme appears to be different concerning SST, with reconstructed SST anomalies of 0.9 to 3.7°C. Due to these relatively low SSS/high SST (and associated changes in stratification), deep convection was apparently strongly reduced in the Nordic Seas during GS, in agreement with all previous studies (e.g. Rasmussen et al., 1996a,b, 1999; Kissel et al., 1999; Van Kreveld et al., 2000; Ballini et al., 2006; Rasmussen and Thomsen, 2009).

Hence, in our MIS3 case, deep convection was likely weaker than during the 119-115 ka BP interval as described in Born et al. (2011). The northward baroclinic volume transport by the NwAC could also be lower in our MIS3 case, even though the northward barotropic transport could have increased, with compensation through a larger export at Fram Strait for instance. Furthermore, subsurface temperature anomalies transported from the south (cf. Fig. 1.d) could imply a larger heat transport due to these warm anomalies, counteracting a potential weaker volume transport. Finally, the larger insolation forcing (insolation at 65°N of ~ 490-500 W/m² during the 30-48 ka BP interval versus ~ 440 W/m² during the 119-115 ka BP interval) could further enhance the impact of this northward heat transport on Nordic Seas SST

We agree that we need to better discuss these different hypotheses in the corrected manuscript.

4- Lines 193–203: I think the discussion that enhanced contribution of moisture from the Norwegian Sea towards Greenland (inferred from SST reconstructions) may played a role in the increase in the deuterium excess recorded in Greenland ice cores during stadials......has discussed in Wary et al. 2016. If so, please summarize and add Wary et al., 2016 as a reference.

The discussion about that in Wary et al. (2016) is quite less explicit, it only concerns the 41-35 ka BP interval, and Greenland deuterium excess data are not directly compared with SST reconstructions. So, if RC2 and the Editor agree with, we would prefer keeping this part in its actual form.

5- Minor issues:
- Please make sure that Müller and Stein, 2014 is included in the reference list.
Sorry about that, we will add it for sure.
- Supporting Information (Line 8): the authors may consider the use of shallow subsurface reservoir age estimates from the northern North Atlantic (e.g., Stern and Lisiecki, 2013; Thornalley et al., 2011) and from the Norwegian Sea (Ezat et al., 2016; Thornalley et al., 2015) to correct for past changes in reservoir ages.
We chose to simply use the Calib 7 automatic reservoir age correction for several reasons:
- Reservoir ages very likely differ depending on the areas and their oceanographical context (in terms of oceanic circulation patterns, stratification/mixing/convection, influence from proximal ice-sheets, other freshwater inputs, …). Hence using reservoir age corrections established in other areas (northern North Atlantic) might not be accurate.
- Reservoir ages have also very likely changed with, and within, the millennial climatic events. Hence, accurate corrections for reservoir age changes implies to use high temporal resolution data of reservoir age variations available for all millennial events of the 40-10 ka BP interval. Such data are available in the North Atlantic sector (Stern and Lisiecki, 2013), but unfortunately only for the last deglaciation in the Norwegian Sea (Thornalley et al., 2015; Ezat et al., 2017).
- Using reservoir age corrections would not have changed anything for the present paper, where age models are firstly constrained with magnetic susceptibility-$\delta^{18}$O NGRIP tie-points on the time interval discussed.

References:
Ballini, M., Kissel, C., Colin, C. and Richter, T.: Deep-water mass source and dynamic associated with rapid climatic variations during the last glacial stage in the North Atlantic: a multi-proxy investigation of the detrital fraction of deep-sea sediments., Geochem Geophys Geosystems, 7(Q02N01), doi:10.1029/2005GC001070, 2006.
Born, A., Nisancioglu, K. H. and Risebrobakken, B.: Late Eemian warming in the Nordic Seas as seen in proxy data and climate models, Paleoceanography, 26(2), PA2207, doi:10.1029/2010PA002027, 2011.
de Vernal, A., Rosell-Melé, A., Kucera, M., Hillaire-Marcel, C., Eynaud, F., Weinelt, M., Dokken, T., Kageyama, M., 2006. Comparing proxies for the reconstruction of LGM sea-surface conditions in the northern North Atlantic. Quaternary Science Reviews 25, 2820–2834.
Ezat, M. M., Rasmussen, T. L. and Groeneveld, J.: Reconstruction of hydrographic changes in the southern Norwegian Sea during the past 135 kyr and the impact of different foraminiferal Mg/Ca cleaning protocols, 2016.
Ezat, M. M., Rasmussen, T. L., Thornalley, D. J. R., Olsen, J., Skinner, L. C., Hönisch, B. and Groeneveld, J.: Ventilation history of Nordic Seas overflows during the last (de)glacial period revealed by species-specific benthic foraminiferal 14C dates, Paleoceanography, 32(2), 2016PA003053, doi:10.1002/2016PA003053, 2017.
Hoff, U., Rasmussen, T. L., Stein, R., Ezat, M. M. and Fahl, K.: Sea ice and millennial-scale climate variability in the Nordic seas 90 kyr ago to present, Nat. Commun., 7, doi:10.1038/ncomms12247, 2016.
Kissel, C., Laj, C., Labeyrie, L., Dokken, T., Voelker, A. and Blamart, D.: Rapid climatic variations during marine isotopic stage 3: Magnetic analysis of sediments from Nordic Seas and North Atlantic, Earth Planet. Sci. Lett., 171(3), 489–502, doi:10.1016/S0012-821X(99)00162-4, 1999.

Müller, J. and Stein, R.: High-resolution record of late glacial and deglacial sea ice changes in Fram Strait corroborates ice–ocean interactions during abrupt climate shifts, Earth Planet. Sci. Lett., 403, 446–455, 2014.

Peck, V. L., Hall, I. R., Zahn, R. and Elderfield, H.: Millennial-scale surface and subsurface paleothermometry from the northeast Atlantic, 55-8 ka BP, Paleoceanography, 23(3), doi:10.1029/2008PA001631, 2008.

Rasmussen, T. L. and Thomsen, E.: Ventilation changes in intermediate water on millennial time scales in the SE Nordic seas, 65-14 kyr BP, Geophys. Res. Lett., 36(1), doi:10.1029/2008GL036563, 2009.

Rasmussen, T. L., Thomsen, E., Labeyrie, L. and Van Weering, T. C. E.: Circulation changes in the Faeroe-Shetland Channel correlating with cold events during the last glacial period (58-10 ka), Geology, 24(10), 937–940, doi:10.1130/0091-7613(1996)024<0937:CCITFS>2.3.CO;2, 1996a.

Rasmussen, T. L., Thomsen, E., Van Weering, T. C. E. and Labeyrie, L.: Rapid changes in surface and deep water conditions at the Faeroe Margin during the last 58,000 years, Paleoceanography, 11(6), 757–771, 1996b.

Rasmussen, T. L., Van Weering, T. C. E. and Labeyrie, L.: Climatic instability, ice sheets and ocean dynamics at high northern latitudes during the last glacial period (58-10 KA BP), Quat. Sci. Rev., 16(1), 71–80, 1997.

Rasmussen, T. L., Balbon, E., Thomsen, E., Labeyrie, L. and Van Weering, T. C. E.: Climate records and changes in deep outflow from the Norwegian Sea ~ 150-55 ka, Terra Nova, 11(2–3), 61–66, 1999.

Stern, J. V. and Lisiecki, L. E.: North Atlantic circulation and reservoir age changes over the past 41,000 years, Geophys. Res. Lett., 40(14), 3693–3697, 2013.

Thornalley, D. J. R., Bauch, H. A., Gebbie, G., Guo, W., Ziegler, M., Bernasconi, S. M., Barker, S., Skinner, L. C. and Yu, J.: A warm and poorly ventilated deep Arctic Mediterranean during the last glacial period, Science, 349(6249), 706–710, doi:10.1126/science.aaa9554, 2015.

Wary, M.: Rôle des conditions océaniques et des ice-shelves en périphérie des calottes européennes lors des évènements climatiques abrupts de la dernière période glaciaire, PhD thesis, Université de Bordeaux. [online] Available from: www.theses.fr/2015BORD0316/document, 2015.

Wary, M., Eynaud, F., Rossignol, L., Lapuyade, J., Gasparotto, M.-C., Londeix, L., Malaizé, B., Castéra, M.-H. and Charlier, K.: Norwegian Sea warm pulses during Dansgaard-Oeschger stadials: Zooming in on these anomalies over the 35–41 ka cal BP interval and their impacts on proximal European ice-sheet dynamics, Quat. Sci. Rev., 151, 255–272, doi:http://dx.doi.org/10.1016/j.quascirev.2016.09.011, 2016.

---

## Author Comment (AC3) · 18 Apr 2017

RC3: Wary et al present reconstruction of surface water conditions in the North Atlantic region during stage three, mainly derived from dinocyst assemblages using transfer functions. The results are compared with climate model hosing experiments and show evidence of a inverse relationship between temperatures in the Nordic Seas and the North Atlantic Ocean/Greenland. The manuscript is well written but I there are some issues that need to be addressed before it should be accepted.

Reply: We thank Anonymous Referee #3 for his/her careful review of our paper. We will take into account all his/her precious comments for the revised version of the manuscript. Below are our replies.

When I read a manuscript that uses transfer functions, I like to start with the raw assemblage data. I was disappointed that the mansuscript does not include these, but I found data for two of the four cores examined in Eynaud et al (2002) and partial assemblage data for a further core in Wary et al (2016).

This is true that most of the data are already presented in published papers: MD95-2009 and MD95-2010 data in Eynaud et al. (2002), MD99-2281 data in Zumaque et al. (2012), and MD99-2285 partial data (35-41 ka BP) in Wary et al. (2016). Complete data for MD99-2285 core should published soon (Wary et al., The southern Norwegian Sea during the last 45 ka: hydrographical reorganizations under changing ice-sheet dynamics, Journal of Quaternary Science, in press). All these papers mention where these data are available and can be downloaded. We did not present the raw data here (except for the relative percentage of *Islandinium minutum*) because of that, but we can for sure add them in Supporting Information and/or indicate the repository where they are available.

Although both of these papers are cited for information about foraminifera assemblages and chronologies, neither is apparently cited for the dinocyst assemblage data. Both these publication also include transfer function derived estimates of sea surface conditions and make similar findings to the present manuscript. The lack of citations to this earlier, overlapping work makes the present manuscript appear more novel than is justified: this must be rectified by citing the authors' previous work appropriately and explicitly stating which parts of the proxy data in the present manuscript are new. The dinocyst stratigraphies that are not already published should be included in the supplementary material.

We agree and will transfer that information from the Supporting Information to the main text.

From the dinocyst assemblages, the manuscript reconstructs summer and winter sea surface temperature and salinities, and sea ice duration using transfer functions. Seasonality is inferred from the difference between summer and winter temperatures. The reported transfer function performances are all impressive, however, these are leave-one-out estimates which, as has been shown repeatedly (Telford 2006; Telford & Birks 2005, 2009, 2011), severely underestimate the true uncertainty in the reconstruction. This is because the environmental variables in the dinocyst calibration set are spatially autocorrelated, violating one of the basic

assumptions of transfer functions (Birks et al 2010). There are cross-validation schemes that are more robust to spatial autocorrelation (Trachsel and Telford 2016): performance statistics from these should be used instead.

It is likely that with a robust cross-validation scheme, the transfer function performance statistics will appear worse and some variables will have little or no skill. I suspect that salinity models are the weakest and that it will be difficult to make independent reconstructions of sea ice duration or winter temperature as both have strong non-linear relationships with summer temperature. Without knowing how large the uncertainty is, the reader cannot evaluate how meaningful the stadial-interstadial difference temperature is.

Repeated parallel studies, also using cross-validation schemes, have also shown that "Although strong spatial autocorrelation characterizes the original climate parameter distribution, the results show that the spatial structure of data has relatively low effect on the calculation of the error of prediction" (Guiot and de Vernal, 2011a, citation from their abstract), concluding that "until a higher performance transfer function approach is developed […] we can only encourage the paleoclimate community to continue using MAT, with all statistical precautions required, as it has been successful for the documenting recent Earth's climate dynamics" (Guiot and de Vernal, 2011b, citation from their conclusion). These studies concerned sea-surface temperature and salinity reconstructions derived from the application of MAT transfer function to dinocyst assemblages. Later, other studies also based on validation exercises showed again the prediction power of the method for sea-ice reconstructions (see Quaternary Science Reviews special issue 79 (2013), especially de Vernal et al., 2013a,b).

As the application of MAT transfer functions to dinocyst assemblages has already been the subject of discussion in CPD, our reply here only provides the main conclusions of works testifying of the reliability and robustness of this method; we refer to the reader to this previous discussion (Milzer et al., 2014: http://www.clim-past.net/10/305/2014/cp-10-305-2014-discussion.html) for further details.

Neither the manuscript nor the precursor papers include any reconstruction diagnostics, such as distance to nearest analogue, which would help the reader evaluate whether the reconstructions can be relied upon.

We can for sure add these data in Supporting Information.

The manuscript needs to make the inclusion criteria for the hosing models explicit. Swingedouw et al (2013) includes six models, but only five are used now. The omitted model is BCM2, which has the opposite temperature response in the Nordic Seas to the other models.

We will move that discussion from the Supporting Information section to the main text.

The combination of the warm dinocyst-inferred surface temperatures and cold planktic foraminifera inferred sub-surface temperatures in the stadials raise some questions. Firstly, why do sub-polar planktic foraminifera not inhabit the surface layer.

RC2 raised a similar same point. We suggested that the subpolar planktonic foraminifera did not inhabit the surface layer because SSS were apparently too low (summer $SSS_{dino}$ between 30.9 and 31.6 on average) according to these species tolerances (e.g., Tolderlund and Bé 1971). We will add some discussion about that in the revised version of our manuscript even if it is already extensively discussed in our next JQS contribution.

Secondly, do the models suggest such a thin surface layer.

According to our data and interpretations, the maximum thickness of this surface layer should be less than ~ 250 m water depth, i.e. the maximum depth of *N. pachyderma s.* habitat reported for this area (e.g. Simstich et al., 2003). In the model simulations, the thickness of the surface layer is quite variable between the different individual simulations (see Figure 9 in Swingedouw et al., 2013, enclosed below), varying from ~ 150 m water depth (HadCM3, IPSLCM5) to ~ 750 m (MPI-ESM, EC-Earth), which allows our hypotheses.

[Figure]

Fig. 9 Latitude-depth section of the temperature in the Atlantic (averaged over 40°W–0°W) for the difference between hosing and control experiments averaged over the 4th decade (unit °C). The *contour* interval is 0.1 °C. a HadCM3, b IPSLCM5, c MPI-ESM, d EC-Earth, e BCM2 and f ORCA05

*Figure 9 in Swingedouw et al., 2013*

**Minor points**
Tables 2 and S4 claim to present anomalies, but appear to be the actual reconstructions.
If they are anomalies, the baseline needs to be specified.
The term anomaly (GS conditions minus GI conditions) is here used to echo the simulation anomalies (hosing experiments minus control experiments), as the aim is to directly confront the reaction of the system between hosing experiment and mean GS conditions, relatively to control experiment and mean GI conditions taken as respective baselines. We can alternatively use the term "differences" for both.

References:

de Vernal, A., Hillaire-Marcel, C., Rochon, A., Fréchette, B., Henry, M., Solignac, S., Bonnet, S., 2013a. Dinocyst-based reconstructions of sea ice cover concentration during the Holocene in the Arctic Ocean, the northern North Atlantic Ocean and its adjacent seas. Quaternary Science Reviews 79, 111–121. doi:10.1016/j.quascirev.2013.07.006

de Vernal, A., Rochon, A., Fréchette, B., Henry, M., Radi, T., Solignac, S., 2013b. Reconstructing past sea ice cover of the Northern Hemisphere from dinocyst assemblages: Status of the approach. Quaternary Science Reviews 79, 122–134.

Eynaud, F., Turon, J.L., Matthiessen, J., Kissel, C., Peypouquet, J.P., De Vernal, A., Henry, M., 2002. Norwegian sea-surface palaeoenvironments of marine oxygen-isotope stage 3: The paradoxical response of dinoflagellate cysts. Journal of Quaternary Science 17, 349–359. doi:10.1002/jqs.676

Guiot, J., de Vernal, A., 2011a. Is spatial autocorrelation introducing biases in the apparent accuracy of paleoclimatic reconstructions? Quaternary Science Reviews 30, 1965–1972. doi:10.1016/j.quascirev.2011.04.022

Guiot, J., de Vernal, A., 2011b. QSR Correspondence "Is spatial autocorrelation introducing biases in the apparent accuracy of palaeoclimatic reconstructions?" Reply to Telford and Birks. Quaternary Science Reviews 30, 3214–3216. doi:10.1016/j.quascirev.2011.07.023

Milzer, G., Giraudeau, J., Schmidt, S., Eynaud, F., and Faust, J.: Qualitative and quantitative reconstructions of surface water characteristics and recent hydrographical changes in the Trondheimsfjord, central Norway, Clim. Past, 10, 305-323, doi:10.5194/cp-10-305-2014, 2014.

Simstich, J., Sarnthein, M., Erlenkeuser, H., 2003. Paired $\delta 18O$ signals of Neogloboquadrina pachyderma (s) and Turborotalita quinqueloba show thermal stratification structure in Nordic Seas. Marine Micropaleontology 48, 107–125.

Swingedouw, D., Rodehacke, C.B., Behrens, E., Menary, M., Olsen, S.M., Gao, Y., Mikolajewicz, U., Mignot, J., Biastoch, A., 2013. Decadal fingerprints of freshwater discharge around Greenland in a multi-model ensemble. Climate Dynamics 41, 695–720. doi:10.1007/s00382-012-1479-9

Tolderlund, D.S., Bé, A.W.H., 1971. Seasonal Distribution of Planktonic Foraminifera in the Western North Atlantic. Micropaleontology 17, 297–329. doi:10.2307/1485143

Wary, M., Eynaud, F., Rossignol, L., Lapuyade, J., Gasparotto, M.-C., Londeix, L., Malaizé, B., Castéra, M.-H., Charlier, K., 2016. Norwegian Sea warm pulses during Dansgaard-Oeschger stadials: Zooming in on these anomalies over the 35–41 ka cal BP interval and their impacts on proximal European ice-sheet dynamics. Quaternary Science Reviews 151, 255–272. doi:http://dx.doi.org/10.1016/j.quascirev.2016.09.011

Wary, M., Eynaud, F., Zaragosi, S., Rossignol, L., Sabine, M., Castéra, M.-H., Billy, I., In press. The Southern Norwegian Sea during the last 45 ka: hydrographical reorganizations under changing ice-sheet dynamics. Journal of Quaternary Science.

Zumaque, J., Eynaud, F., Zaragosi, S., Marret, F., Matsuzaki, K.M., Kissel, C., Roche, D.M., Malaizé, B., Michel, E., Billy, I., Richter, T., Palis, E., 2012. An ocean–ice coupled response during the last glacial: a view from a marine isotopic stage 3 record south of the Faeroe Shetland Gateway. Climate of the Past 8, 1997–2017.

---

## Author Response (AR1)

Dear editor,

Please find below our previous detailed replies to referees' comments (in blue), our reply to your comments (in red), and the associated modifications of the original manuscript (also in red).

Attached to it are (i) the "normal" revised version of the manuscript, and (ii) the same revised version of the manuscript where modifications done are highlighted in colors.

We hope we successfully complied with the whole remarks and managed to improve the quality of our paper.

Yours sincerely,

Mélanie Wary

**Reply to your comments**

Dear authors,

Your paper was seen by 3 reviewers and all suggested minor revisions, the more extensive coming from reviewers 2 and 3. The points raised are appropriate and useful. On the whole your response and suggested revisions are suitable. Therefore, I encourage you to submit a revised version of the manuscript considering the reviewers suggestions.

In your reply to the comments of reviewer 2, you provided answers to their queries but did not specify if you would be including this information in your revised manuscript; I would urge you to include much of this in an abbreviated form.

We included answers in the text most of these queries (see our comments and the changes we made in the section related to referee 2's comments).

As requested, I think it is ok to keep L193-203 as it stands, but you could add Wary et al 2016 as a reference.

Reference added (lines 219-220).

Reviewer 3 raised some very significant issues regarding the validity of dinocyst assemblage techniques to independantly reconstruct multiple oceanic parameters. Your revised version must address and include these. Your original manuscript did not present the reader with an adequate representation of the state of the literature on this topic, namely that the issue of autocorrelation for dinocyst assemblages has been heavily debated and certainly remains an issue, albeit one that is contentious amongst different communities. Examining the data presented in your orginal manuscript (fig 2), the changes in SST appear very similar to the inferred sea-ice changes in each core, and therefore this topic seems very relevant to your dataset. In your revised version, the issues raised by reviewer 3 need to be fully acknowledged and you should also include reviewer 3's suggested references and the inferred uncertainty on reconstructed parameters provided by those studies, in addition to your orginal error estimates. This will ensure that readers are fully informed on (and can research) this topic and reach their own conclusions regarding the robustness of your methodology.

We now discuss this in the Supporting Information Section S2 lines 82-95 (where we also cite reviewer 3's suggested references). We also added the distance to the nearest analogue in the Supporting Information Fig. S5. However, we cannot include the RMSEP calculated in these studies since they are not related to the same database that the one we use here (940 *versus* 1207 datapoints).

In your response to reviewer 3's question about modelled surface layer depths, your answer suggested modelled depths of 150-750m were shallow enough to be deeper than the habitat of N. pachyderma, which may be as deep as 250m in the Nordic Seas. However, Simstich et al suggest 70-130m for the modern Nordic Seas influenced by cold Arctic water, which is lperhaps a closer analogue to your inferred stadial conditions. And also what about bulloides and quinqueloba which have much shallower depth? In essence, the modelled surface layers do not seem not fully compatible with your data; this is not detrimental, but worth noting.

Actually, in the Arctic, which is potentially another close analogue (relatively "warm" and fresh surface layer seasonally covered with sea-ice, and separated by a halocline from a colder and saltier subsurface layer), *N. pachyderma* habitat has been reported even deeper than 250 m (see for example Volkmann and Mensch, 2001 – reference cited in the Supporting Information Section S6 line 160 – or Hillaire-Marcel and de Vernal, 2008, cited in the main text line 211). Furthermore, our model simulations depict anomalies, comparable in our case to GI-GS differences. According to our subsurface data, there is no apparent subsurface GI-GS differences, and according to other subsurface data form the Nordic Seas data (e.g. Rasmussen and Thomsen, 2004; Dokken et al., 2013; Wary et al., 2016) there is even a slight warming with subsurface temperatures staying < c. 6°C (i.e. subsurface temperatures still well below our reconstructed SST). Concerning *G. bulloides* and *T. quinqueloba*, as mentioned in our replies to the referees' comments, SSS are too low (both in winter and summer, both during GI and GS, cf. Table S4) to enable their development in the surface layer they usually inhabit. Hence, we keep on thinking that our model results and our reconstructions are compatible concerning this surface layer.
We did not add any mention about that, but we can do it if you think it is necessary.

Finally, some additional comments from my own reading of the manuscript: In the introduction section you need to explicitly discuss the concept that previous foram based studies have inferred incursions of warm subsurface water during stadials (Dokken et al 2013 and Rasmussen et al 2004). As it stands, the introduction suggests foram based studies have only reconstructed cold conditions in the Nordic Seas; this is not the case.

This is now discussed lines 40-43.

Line 109 - please can you include any statistics to support this? I am not wholly convinced of this relationship, especially for MD95-2009 and -2010

We included a new table (Table 3, line 505, referred to lines 117 and 123) presenting the correlation coefficients between NGRIP $\delta^{18}O$ and the winter SST reconstructions of the four cores. As we previously mentioned, for cores MD99-2285 and MD95-2009 "SST is systematically anti-correlated against Greenland and North Atlantic temperatures" (line 122) as supported by these statistics; this is not so clear for core MD95-2010 due to lower resolution and sensitivity, but this scheme is here (and in the other cores) supported by our raw dinocyst assemblages (see lines 125-128 and Figs. S2 and S3).

Line 154 - your findings do not relate to the whole/wider Nordic Seas but are restricted to eastern coastal sites; please reword accordingly.

We replaced "Nordic Seas" by "Norwegian Sea", now line 172.

Thank you for submitting your interesting manuscript and I look forward to receiving the revised version, addressing the above points.

Yours sincerely,

David Thornalley

REV#1: This paper presents interesting reconstructions of sea-surface conditions in the North Atlantic and Nordic Seas for Marine Isotopic Stage 3, based on dinocyst assemblages and planktonic forams, as well as climate modelling. This is a very well written paper, well presented and argued, with little to fault. The only aspect I would like to have seen being discussed is the possible forcing of productivity on dinocyst assemblages, in particular the high abundance of I. minutum, which is recognized as a tracer of sea-ice cover, but also abundant in high nutrient environments (see Zonneveld et al 2013). Further studies by Heikkilä et al (2014, 2016) also suggest a more complex response of this species to sea-ice environments. Based on these ecological findings, how would it affect your interpretation?

Heikkilä, M., Pospelova, V., Forest, A., Stern, G.A., Fortier, L., Macdonald, R.W. Dinoflagellate cyst production over an annual cycle in seasonally ice-covered Hudson Bay (2016) Marine Micropaleontology, 125, pp. 1-24
Heikkilä, M., Pospelova, V., Hochheim, K.P., Kuzyk, Z.Z.A., Stern, G.A., Barber, D.G., Macdonald, R.W. Surface sediment dinoflagellate cysts from the Hudson Bay system and their relation to freshwater and nutrient cycling (2014)

We are grateful to reviewer 1 for his / her review of our paper and for pointing out the interesting role of productivity on dinocyst assemblages in sea-ice covered environments.

The highest abundances of *I. minutum*, and especially abundances as high as those recorded during GI in our Norwegian Sea cores, are systematically encountered in cold and sea-ice covered environments (Figure S2, re-enclosed below). Nonetheless, in these areas, this heterotrophic taxon can exhibit a complex spatial and temporal dynamic tightly linked to nutrient and prey availability, as highlighted by Heikkilä et al. (2014, 2016) for the Hudson Bay and Hudson Strait where this factor appears as the primary controlling factor. We will for sure add a mention about that in the Supporting Information Section S2.
Mention added in Supporting Information Section S2 lines 46-48, and associated references added in the main text lines 128.

However, in our case, we think that sea ice, stratification and nutrient/prey availability changes are directly related to each other and play in concert.
During GS, our dinocyst assemblages closely resemble those of the Eastern Hudson Bay as described by Heikkilä et al. (2014; i.e. *P. dalei*, *O. centrocarpum*, *S. ramosus*), with the exception of *B. tepikiense* being additionally reported in our Norwegian Sea cores and in significant proportions (see Eynaud et al., 2002 and Wary et al., in press, The southern Norwegian Sea during the last 45 ka: hydrographical reorganizations under changing ice-sheet dynamics. Journal of Quaternary Science). Heikkilä et al. (2014) related this assemblage to productive stratified waters related to large meltwater inputs and a relatively long open-water season. These findings are in agreement with our interpretations for GS. The occurrence of *B. tepikiense* in our assemblages reinforces the stratification pattern (together with a strongest seasonality), and dinocyst-derived (through MAT transfer function) primary productivity reconstructions for core MD99-2285 (see Fig. 6 in Wary et al., 2016, enclosed below) support the high productivity pattern.

During GI, our dinocyst assemblages are dominated by heterotrophic taxa, with the strong dominance of *I. minutum* ('sea-ice indicator') and the lesser occurrence of *Brigantedinium* spp. ('nutrient indicator') in cores MD95-2009 and MD95-2010 (Eynaud et al., 2002). Heterotrophic taxa usually feed on diatoms, and Hoff et al. (2016) indeed reported higher diatom fluxes during GI. If not related to a better preservation effect (less dissolution), this could indicate more favorable conditions for diatom proliferation, and especially in the present case likely for sea-ice diatoms proliferation (higher IP25 abundances are indeed reported during GI in core MD99-2285, see Wary, 2015). Our records indicate the absence of stratification during GI (with longer sea-ice cover limiting iceberg calving and subsequent meltwater inputs, see Wary et al., 2016 and references therein), which is reported by Heikillä et al. (2014) in the Hudson Strait as a factor favoring diatom proliferation and disfavoring competitive autotrophic dinoflagellate development. Hence during GI, sea-ice, stratification and nutrient/prey availability appear directly related to each other, and could all together provide optimal conditions for *I. minutum* and *Brigantedinium* spp. seasonal proliferation: longer sea-ice cover durations, colder SST, likely less (compared to GS) but still substantial nutrient-rich meltwater inputs (from the seasonal melting of sea-ice and from continental freshwater inputs likely enhanced under warmer atmospheric conditions), less stratification (less iceberg melting), more (sea-ice) diatoms/i.e. heterotrophic dinocyst preys, less autotrophic dinoflagellates, … and more heterotrophic taxa typical of cold, seasonally ice covered, nutrient- and sea-ice diatom-rich, but low primary productivity (likely due to grazing; see Fig. 6 in Wary et al., 2016) environments.

**Figure S2 (Supporting Information from the present reviewed paper):**

[Figure]

[Figure]

[Figure]

*Figure S2. Islandinium minutum distribution and ecology. (a) Islandinium minutum distribution within the modern dinocyst database made of 1207 points. (b) Oceanic temperatures at 10 mbsl (WOA09 data; Locarnini et al., 2010). (c) Sea-ice cover (with concentration greater than 50%) duration within the modern dinocyst database made of 1207 points (after data provided by the National Climate Data Centre in Boulder). These maps demonstrate the strong link of this dinocyst taxon with cold and seasonally sea-ice covered surface environments.*

[Figure]

Fig. 6. Interpretation of NPS absolute abundance signal in core MD99-2285. (a) NGRIP d18O regional stratotype. (b) Dinocysts-derived mean annual primary productivity. (c) NPS relative abundance (plotted with a reverse scale ranging from 94 to 100%). (d) NPS absolute abundance, compared with B. tepikiense relative abundance. (e) Total planktonic foraminifera absolute concentration. (f) Total dinocyst absolute concentration. Hatched bands highlight stadial intervals (age limits after Wolff et al., 2010).

**Reply** to the **Interactive comment on "Regional seesaw between North Atlantic and Nordic Seas during the last glacial abrupt climate events" by Mélanie Wary et al., by Anonymous Referee #2,

RC2: Wary et al present an interesting compilation of (and new) sea surface temperature, salinity, sea ice cover reconstructions from the Norwegian Sea and northern North Atlantic based on dinocyst analyses for Marine Isotope Stage 3 as well as an ensemble of freshwater hosing experiments run under preindustrial boundary conditions. The paper is well written and data are very well presented and adds to the debate about the stadial/interstadial evolution of the Nordic Seas circulation during the last glacial and its role in the abrupt climate change. However, the paper needs moderate/major revisions before it could be accepted for publication.

Reply: We want to thank Anonymous Referee #2 for his/her careful and constructive review of our paper. We will take into account all his/her precious advice for the revision of the manuscript. Below are our replies to his/her comments.

1- First of all the authors need to elaborate on how summer SST up to 14°C in the Norwegian Sea during stadials compare with other previous reconstructions. In this regard, the following points need further discussion:

- The authors stated 'Furthermore, the few direct but qualitative sea-ice reconstructions based on lipid biomarker analyses (Müller and Stein, 2014; Hoff et al., 2016) yielded contrasting results'. Looking at Figure 4 in Hoff et al 2016, it does not seem that those two studies are at odd. In contrary, the sea ice cover records in Müller and Stein, 2014 and Hoff et al., 2016 seem for me to correlate well.

We totally agree, this figure (enclosed below) clearly shows the same results, i.e. low PIP25 values during Heinrich event 1 (and likely Heinrich event 2 in-between GI 3 and 2 – "IS3" and "IS2" in this figure). However, the interpretations of authors significantly differ:

-   Müller and Stein (2014) interpreted the sharp decrease down to very low PIP25 values at the beginning of HE1 as the "**sudden break-up of the ice cover and the concentrated release of high amounts of sea ice and icebergs trapped therein**" (page 453, Section 6).
-   In Hoff et al. (2016), as Heinrich event 1 is "characterized by IP25 and brassicasterol (as well as dinosterol) values of close to or zero", "PBIP25 and PDIP25 values of 1 were assumed" (page 4, legend Figure 4) and interpreted by the authors as "**a very prominent period of perennial or near-perennial sea ice cover**" (page 6).

The use of the word "results" was for sure inappropriate, and we will replace it by "interpretations".

The word "results" has been replaced by "interpretations", line 45.

[Figure]

**Figure 4 | Sea ice proxies for the time interval between 30 and 8 kyr ago.** Comparison of $P_BIP_{25}$ and $P_DIP_{25}$ signals of sediment core JM11-FI-19PC (GICC05-timescale) and sediment core MSM5/5-712-2 (calibrated age scale) from the Svalbard margin published by Müller and Stein[17]. For H-event 1 (blue bar), characterized by $IP_{25}$ and brassicasterol (as well as dinosterol) values of close to or zero (that is, 'zero divided by zero'), per definition maximum $P_BIP_{25}$ and $P_DIP_{25}$ values of 1 were assumed (for background see refs 15,26). For core locations, see Fig. 1. BA, Bølling and Allerød interstadial; H, H-event; HS, Heinrich stadial; LGM, Last Glacial maximum; YD, Younger Dryas.

*Figure 4 from Hoff et al. (2016)*

I think it is critical to discuss why sea ice cover reconstructions in the southern Norwegian Sea in this study (dinocyst-based) and in Hoff et al., 2016 (lipid biomarker- based) significantly differ. I suggest you plot IP25, brassicasterol- and dinosterol concentration (not the PBIP25 and PDIP25 indexes) with your dinocyst-based data and see if you can reconcile between them or at least make the apparent disagreement between the two results clear, so future investigations may take it further.

We also think it is critical to understand why our dinocyst-based sea-ice reconstructions and Hoff et al. (2016) biomarker-based sea-ice reconstructions significantly differ. But to do that properly, we think it is first needed to clarify why the same biomarker-derived signals lead to opposite interpretations (Hoff et al., 2016 *versus* Müller and Stein, 2014), and why the same biomarker proxy (IP25 abundances) can lead to opposite results in the same studied area (Hoff et al., 2016 on core JM11-FI-19PC *versus* Sicre, unpublished data on core MD99-2285, see figure p.194 of Wary, 2015: www.theses.fr/2015BORD0316/document). Unfortunately, we are not allowed to provide you with the plot of these MD99-2285 IP25 data along with Hoff et al. (2016)'s data, and we are truly sorry about that. But if you look at Figure 4b on p.194 of Wary (2015) and compare with Hoff et al. (2016)'s IP25 abundances (in µg/g of dry sed. for more coherency) you can see that both signals exhibit similar trends on the 39-36 ka BP section (i.e., higher IP25 abundances during GI8 and at the end of HS4, and lower ones during GS8), but rather different ones on the 41-39 ka BP interval (i.e. for core MD99-FI-19PC), lower (higher) IP25 abundances during GS10 (i.e. the "H4" IP25 peak in Hoff et al., 2016), higher (lower) IP25 abundances during GI9, and zero (low/moderate) IP25 abundances during HE4 interval as defined on the basis of our IRD data). In core MD99-2285, IP25 measurements were realized at very high temporal resolution (54 years on average) and they reveal a very noisy signal (but yet with clear trends), so the difference might maybe (or maybe not) come from this given the lower resolution in Hoff et al. (2016). But, whatever the reason(s) might be, we think that we are not the most appropriate co-author team to discuss that (no biomarker people among us), that our paper is not the most appropriate to discuss that (no biomarker data), and thus that it would be quite improper to discuss such discrepancies in our paper.

The difference in interpretation is mentioned in the text line 43-45.  Concerning the opposition between results from two nearby cores (Hoff et al., 2016 *versus* unpublished data form core MD99-2285), if you agree, we would prefer not mentioning it, because as we said our team and our paper are not the most appropriate to discuss that, and most of all because we are not allowed to use these

IP25 data from core MD99-2285, and even their mention in a published paper is a sensitive issue (it has been the case for our QSR paper).

- The authors may need to explain why the %subpolar planktic foraminifera (e.g., T. quinqueloba and G. bulloides) did not increase during stadials if summer SST was that high in the Norwegian Sea. I think the conditions at the average calcification depth of N. pachyderma may be best recorded in the isotopic and elemental composition of its shells, whereas the % N. pachyderma is also controlled by the abundance of other planktic species. For example, Mg/Ca in N. pachyderma shows different pattern from % N. pachyderma for Heinrich Stadial 1, also in the southern Norwegian Sea (Ezat et al., 2016).

We suggest that the % subpolar planktonic foraminifera did not increase during GS because: (i) in the surface layer, SSS were apparently too low (summer $SSS_{dino}$ between 30.9 and 31.6 on average) according to these species tolerances (e.g., Tolderlund and Bé 1971), and (ii) in the subsurface layer, temperatures were too low (below 5-6°C according to *N. pachyderma* s. optimal temperature range – e.g., Tolderlund and Bé 1971) according to these species tolerances, but these subsurface temperatures could have varied between 1°C and 5-6°C as shown in Ezat et al. (2016) and also suggested in Wary et al. (2016), in both cases for stadial intervals characterized by 100% NPS.

We added some information about that in the main text lines 176-178 and in the Supporting Information Section S5 lines 144-148.

- Notably, the reconstructed summer temperatures during glacial stadials in the southern Norwegian Sea in this study are similar to or even higher than modern temperatures. It is not plausible that we ignore an observation just because it does not fit with what we may expect.

Modern SST values are indicated on Figure S4. We did not indicate them on Figure 2, because the SST range is too low to do so. But we can add these modern SST values on Figure S5 where annual, summer and winter SST are presented, and we can also mention it in the text as we already did in our previous papers.

We included modern seasonal SST values on SI Fig. S5, and added a mention about that in the main text line 130.

Furthermore, comparable warm anomalies (as high as modern temperatures) were obtained (also on the basis of dinocyst reconstructions – but not exclusively – and of a multicore compilation) during the last glacial maximum (see de Vernal et al., 2006 for instance), suggesting an atypical and a much more complex functioning of the GIN seas at that time, quite different from the existing modern polar schemes we have in mind.

We mentioned the fact that our reconstructed SST during GS in the Norwegian were sometimes higher than the modern ones, such as it has also been reported during the LGM in the Nordic Seas, by adding "with also sometimes warmer than modern SST in the Nordic Seas," lines 128-131.

However, more discussion needed regards the temperature at the source of these water, were the stadial temperatures at lower latitudes higher than modern? In addition, the inflowing water may have had to mix with more cold polar water than in the modern case in its way to the Nordic seas.

According to our simulations, the initial freshwater flux induces a mid-latitude subsurface warming due to the meltwater lid, and the water mass transport is increased along the eastern North Atlantic boundary, potentially through the Continental Slope Current (flowing poleward along the European margin) as suggested in Wary et al. (2016). The source of this water is quite enigmatic, even if we consider that it is advected through this near-surface current whose origin is still debated (see Section 2 in Wary et al., 2016). Nonetheless, Peck et al. (2008) reported unexpectedly warm SST (sometimes warmer than modern ones) during Heinrich stadials at the Porcupine Seabight (~51°N) on the basis of Mg/Ca measurements on *G. bulloides* (see their Figure 4, enclosed below). They proposed several hypotheses to explain these too warm SST during Heinrich events, one of them being transient advection of subtropical surface waters through the Continental Slope Current.

We added a mention about that lines 153-156. Reference to Peck et al. (2008) was added in the reference list (lines 397-398).

[Figure]

**Figure 4.** Temperature estimates and seasonality of the upper water column at MD01-2461 over the last 55 ka. (a) GISPII $\delta^{18}$O ice core record. (b) Summer SST inferred from Mg/Ca$_{G.\ bulloides}$ (red solid circles) following *Peck et al.* [2006]. Modern summer SST shown by red dashed line. Mean annual subsurface temperature inferred from Mg/Ca$_{N.\ pachyderma\ sinistral}$ (blue open circles) following *Peck et al.* [2006]. Modern mean annual water temperature at 150 m shown by blue dashed line. (c) June solstice solar insolation at 65°N (La90) [*Laskar*, 1990]. (d) $\Delta T$ (summer subsurface temperature (SST), °C). A 6th-order polynomial fit is shown in red. Modern $\Delta T$ and $\Delta T = 0$ shown with dashed lines.

*Figure 4 from Peck et al. (2008)*

2- It is important to clearly clarify in the methods (in section 2.2) what new data have been generated in this study and what have been used from previous studies. I think most of the dinocyst analyses from cores MD95-2009, MD95-2010 and MD99-2285 are already published, or? I may have just missed the referring to previous studies, so I hope this comment does annoy the authors if that is the case.

MD99-2281, MD95-2009 and MD95-2010 dinocyst assemblages are already published, as well as for core MD99-2285 on the 41-35 ka BP interval. For the present study, the published dinocyst counts of those cores and the new data of core MD99-2285 were used to generate new SST, SSS and SIC estimates thanks to the extended modern database used for the transfer function. We will move these information from the Supporting Information section to Section 2.2.

We added a mention in Section 2.2 to clarify the new data generated in the present study and the previously published data we used (lines 73-77). We also kept this information, with more details and additional information about data from core MD99-2285, in the Supporting Information Section S2 (lines 66-71).

Related to this, I would suggest removing the word 'Surprisingly' from the following sentence "Surprisingly, the three Norwegian Sea cores record higher SST and shorter SIC durations during the cold North Atlantic GS, and lower SST and longer SIC durations during the warm North Atlantic GI." This is not a surprise as previous dinocyst- based studies have already showed this stadial/interstadial SST pattern in the Norwegian Sea (e.g., Eynaud et al., 2002; Wary et al., 2016).

Our formulation is probably awkward, we do not mean that the results are surprising in the sense that they are new, but that the scheme described here at a regional scale is surprising because different from the accepted scheme. We will therefore reformulate our sentence.

"Surprisingly" has been replaced by "Paradoxically", line 118.

3- Line 139-144: Weakening of the subpolar gyre has been employed to explain relative warming in the eastern Nordic Seas under interglacial conditions for example during late Eemian (e.g., Born et al., 2011). However, a key difference here (in addition to many others), is the likely significant suppression of deep water formation in the Nordic Seas during MIS3 stadials, which has an impact of the inflowing surface water. So, adding more lines of discussions here is merited.

In Born et al. (2011) deep water convection remains active in the Nordic Seas during the episode (119-115 ka BP) of weakening of the subpolar gyre circulation / increased heat transport in the Nordic Seas through the Norwegian Atlantic Current (NwAC, i.e. probable continuation of the Continental Slope Current). According to the authors, this is because even if "enhanced salt transport counteracts the general freshening […] locally but does not reverse it" and "despite increased heat transport by the NwAC, no absolute warming of the Nordic Seas is expected at 115 ka because of the counteracting large insolation forcing".

In our MIS3 case, it seems to be the same situation for the surface freshening since our reconstructed GI-GS SSS anomalies are quite small (0.0 to 0.4 psu). However, the scheme appears to be different concerning SST, with reconstructed SST anomalies of 0.9 to 3.7°C. Due to these relatively low SSS/high SST (and associated changes in stratification), deep convection was apparently strongly reduced in the

Nordic Seas during GS, in agreement with all previous studies (e.g. Rasmussen et al., 1996a,b, 1999; Kissel et al., 1999; Van Kreveld et al., 2000; Ballini et al., 2006; Rasmussen and Thomsen, 2009).

Hence, in our MIS3 case, deep convection was likely weaker than during the 119-115 ka BP interval as described in Born et al. (2011). The northward baroclinic volume transport by the NwAC could also be lower in our MIS3 case, even though the northward barotropic transport could have increased, with compensation through a larger export at Fram Strait for instance. Furthermore, subsurface temperature anomalies transported from the south (cf. Fig. 1.d) could imply a larger heat transport due to these warm anomalies, counteracting a potential weaker volume transport. Finally, the larger insolation forcing (insolation at 65°N of ~ 490-500 W/m² during the 30-48 ka BP interval versus ~ 440 W/m² during the 119-115 ka BP interval) could further enhance the impact of this northward heat transport on Nordic Seas SST

We agree that we need to better discuss these different hypotheses in the corrected manuscript.

We now discuss that, in an abbreviated form, lines 156-161.

4- Lines 193–203: I think the discussion that enhanced contribution of moisture from the Norwegian Sea towards Greenland (inferred from SST reconstructions) may played a role in the increase in the deuterium excess recorded in Greenland ice cores during stadials......has discussed in Wary et al. 2016. If so, please summarize and add Wary et al., 2016 as a reference.

The discussion about that in Wary et al. (2016) is quite less explicit, it only concerns the 41-35 ka BP interval, and Greenland deuterium excess data are not directly compared with SST reconstructions. So, if RC2 and the Editor agree with, we would prefer keeping this part in its actual form.

As you agreed and suggested, we kept this part in this original form and added reference to Wary et al. (2016) lines 220-221.

5- Minor issues:

- Please make sure that Müller and Stein, 2014 is included in the reference list.

Sorry about that, we will add it for sure.

Done lines 392-394.

- Supporting Information (Line 8): the authors may consider the use of shallow subsurface reservoir age estimates from the northern North Atlantic (e.g., Stern and Lisiecki, 2013; Thornalley et al., 2011) and from the Norwegian Sea (Ezat et al., 2016; Thornalley et al., 2015) to correct for past changes in reservoir ages.

We chose to simply use the Calib 7 automatic reservoir age correction for several reasons:

- Reservoir ages very likely differ depending on the areas and their oceanographical context (in terms of oceanic circulation patterns, stratification/mixing/convection, influence from proximal ice-sheets, other freshwater inputs, …). Hence using reservoir age corrections established in other areas (northern North Atlantic) might not be accurate.
- Reservoir ages have also very likely changed with, and within, the millennial climatic events. Hence, accurate corrections for reservoir age changes implies to use high temporal resolution data of reservoir age variations available for all millennial events of the 40-10 ka BP interval. Such data are available in the North Atlantic sector (Stern and Lisiecki, 2013), but unfortunately only for the last deglaciation in the Norwegian Sea (Thornalley et al., 2015; Ezat et al., 2017).

- Using reservoir age corrections would not have changed anything for the present paper, where age models are firstly constrained with magnetic susceptibility-$\delta^{18}$O NGRIP tie-points on the time interval discussed.

A mention about this has been added in Supporting Information Section S1 lines 29-35.

Although both of these papers are cited for information about foraminifera assemblages and chronologies, neither is apparently cited for the dinocyst assemblage data. Both these publication also include transfer function derived estimates of sea surface conditions and make similar findings to the present manuscript. The lack of citations to this earlier, overlapping work makes the present manuscript appear more novel than is justified: this must be rectified by citing the authors' previous work appropriately and explicitly stating which parts of the proxy data in the present manuscript are new. The dinocyst stratigraphies that are not already published should be included in the supplementary material.

We agree and will transfer that information from the Supporting Information to the main text.

We did not transfer this information from SI to the main text, but we added some mentions or sentences to clearly specify it both in the main text lines 73-77 and in the SI Section S2 lines 66-71.

From the dinocyst assemblages, the manuscript reconstructs summer and winter sea surface temperature and salinities, and sea ice duration using transfer functions. Seasonality is inferred from the difference between summer and winter temperatures. The reported transfer function performances are all impressive, however, these are leave-one-out estimates which, as has been shown repeatedly (Telford 2006; Telford & Birks 2005, 2009, 2011), severely underestimate the true uncertainty in the reconstruction. This is because the environmental variables in the dinocyst calibration set are spatially autocorrelated, violating one of the basic assumptions of transfer functions (Birks et al 2010). There are cross-validation schemes that are more robust to spatial autocorrelation (Trachsel and Telford 2016): performance statistics from these should be used instead.

It is likely that with a robust cross-validation scheme, the transfer function performance statistics will appear worse and some variables will have little or no skill. I suspect that salinity models are the weakest and that it will be difficult to make independent reconstructions of sea ice duration or winter temperature as both have strong non-linear relationships with summer temperature. Without knowing how large the uncertainty is, the reader cannot evaluate how meaningful the stadial-interstadial difference temperature is.

Repeated parallel studies, also using cross-validation schemes, have also shown that "Although strong spatial autocorrelation characterizes the original climate parameter distribution, the results show that the spatial structure of data has relatively low effect on the calculation of the error of prediction" (Guiot and de Vernal, 2011a, citation from their abstract), concluding that "until a higher performance transfer function approach is developed […] we can only encourage the paleoclimate community to continue using MAT, with all statistical precautions required, as it has been successful for the documenting recent Earth's climate dynamics" (Guiot and de Vernal, 2011b, citation from their conclusion). These studies concerned sea-surface temperature and salinity reconstructions derived from the application of MAT transfer function to dinocyst assemblages. Later, other studies also based on validation exercises showed again the prediction power of the method for sea-ice reconstructions (see Quaternary Science Reviews special issue 79 (2013), especially de Vernal et al., 2013a,b).

As the application of MAT transfer functions to dinocyst assemblages has already been the subject of discussion in CPD, our reply here only provides the main conclusions of works testifying of the reliability and robustness of this method; we refer to the reader to this previous discussion (Milzer et al., 2014: http://www.clim-past.net/10/305/2014/cp-10-305-2014-discussion.html) for further details.

We added a discussion about that In SI Section S2 lines 82-95.

Neither the manuscript nor the precursor papers include any reconstruction diagnostics, such as distance to nearest analogue, which would help the reader evaluate whether the reconstructions can be relied upon.

We can for sure add these data in Supporting Information.

We included these data in SI Fig. S5.

The manuscript needs to make the inclusion criteria for the hosing models explicit. Swingedouw et al (2013) includes six models, but only five are used now. The omitted model is BCM2, which has the opposite temperature response in the Nordic Seas to the other models.

We will move that discussion from the Supporting Information section to the main text.

We did not move it but added a sentence about it in the main text lines 86-88.

The combination of the warm dinocyst-inferred surface temperatures and cold planktic foraminifera inferred sub-surface temperatures in the stadials raise some questions. Firstly, why do sub-polar planktic foraminifera not inhabit the surface layer.

RC2 raised a similar point. We suggested that the subpolar planktonic foraminifera did not inhabit the surface layer because SSS were apparently too low (summer $SSS_{dino}$ between 30.9 and 31.6 on average) according to these species tolerances (e.g., Tolderlund and Bé 1971). We will add some discussion about that in the revised version of our manuscript even if it is already extensively discussed in our next JQS contribution.

As mentioned previously, we added some information about that in the main text lines 176-178 and in the Supporting Information Section S5 lines 144-148.

Secondly, do the models suggest such a thin surface layer.

According to our data and interpretations, the maximum thickness of this surface layer should be less than ~ 250 m water depth, i.e. the maximum depth of *N. pachyderma s*. habitat reported for this area (e.g. Simstich et al., 2003). In the model simulations, the thickness of the surface layer is quite variable between the different individual simulations (see Figure 9 in Swingedouw et al., 2013, enclosed below), varying from ~ 150 m water depth (HadCM3, IPSLCM5) to ~ 750 m (MPI-ESM, EC-Earth), which allows our hypotheses.

As said in our reply to your comment, we did not add any information about that in the revised version, but we can do it if we think it would improve our manuscript.

[Figure]

Fig. 9 Latitude-depth section of the temperature in the Atlantic (averaged over 40°W–0°W) for the difference between hosing and control experiments averaged over the 4th decade (unit °C). The *contour* interval is 0.1 °C. a HadCM3, b IPSLCM5, c MPI-ESM, d EC-Earth, e BCM2 and f ORCA05

*Figure 9 in Swingedouw et al., 2013*

**Minor points**

Tables 2 and S4 claim to present anomalies, but appear to be the actual reconstructions.

If they are anomalies, the baseline needs to be specified.

The term anomaly (GS conditions minus GI conditions) is here used to echo the simulation anomalies (hosing experiments minus control experiments), as the aim is to directly confront the reaction of the system between hosing experiment and mean GS conditions, relatively to control experiment and mean GI conditions taken as respective baselines. We can alternatively use the term "differences" for both.

For more clarity, we finally preferred keeping the term "anomalies" since we clearly define it in the SI Section S4 and since it has been similarly used in previous paleoclimate studies (e.g. Kiefer, T., Lorenz, S., Schulz, M., Lohmann, G., Sarnthein, M., Elderfield, H.: Response of precipitation over

Greenland and the adjacent ocean to North Pacific warm spells during Dansgaard–Oeschger stadials. Terra Nova 14, 295–300. doi:10.1046/j.1365-3121.2002.00420.x, 2002).

**Regional seesaw between North Atlantic and Nordic Seas during the last glacial abrupt climate events**

Mélanie Wary[1], Frédérique Eynaud[1], Didier Swingedouw[1], Valérie Masson-Delmotte[2], Jens Matthiessen[3], Catherine Kissel[2], Jena Zumaque[1,4], Linda Rossignol[1], Jean Jouzel[2]

[1]UMR 5805, EPOC (Environnements et Paléoenvironnements Océaniques et Continentaux), CNRS-EPHE-Université de Bordeaux, 33615 Pessac, France.
[2]UMR8212, LSCE (Laboratoire des Sciences du Climat et de l'Environnement)/IPSL (Institut Pierre Simon Laplace), CEA/CNRS-INSU/UVSQ, 91191 Gif-sur-Yvette CEDEX, France.
[3]AWI (Alfred Wegener Institute), Helmholtz Centre for Polar and Marine Research, 27568 Bremerhaven, Germany.
[4]Now at GEOTOP, UQAM, Montréal, Québec H3C 3P8, Canada.

*Correspondance to:* Mélanie Wary (melanie.wary@u-bordeaux.fr)

**Abstract.** Dansgaard-Oeschger oscillations constitute one of the most enigmatic features of the last glacial cycle. Their cold atmospheric phases have been commonly associated with cold sea-surface temperatures and expansion of sea ice in the North Atlantic and adjacent seas. Here, based on dinocyst analyses from the 48-30 ka BP interval of four sediment cores from the northern Northeast Atlantic and southern Norwegian Sea, we provide direct and quantitative evidence of a regional paradoxical seesaw pattern: cold Greenland and North Atlantic phases coincide with warmer sea-surface conditions and shorter seasonal sea-ice cover durations in the Norwegian Sea as compared to warm phases. Combined with additional paleorecords and multi-model hosing simulations, our results suggest that during cold Greenland phases, reduced Atlantic meridional overturning circulation and cold North Atlantic sea-surface conditions were accompanied by the subsurface propagation of warm Atlantic waters that re-emerged in the Nordic Seas and provided moisture towards Greenland summit.

**1    Introduction**

The last glacial cycle has been punctuated by abrupt climatic variations strongly imprinted in Greenland ice core records where they translate into millennial oscillations between cold (Greenland stadial, GS) and warm (Greenland interstadial, GI) atmospheric phases (e.g., North Greenland Ice Core Project members, 2004). They are tightly linked to pan-North Atlantic ice-sheet dynamic that manifests itself by cyclic iceberg releases concomitant with GS (Bond and Lotti, 1995). These variations are thought to be linked to changes in the North Atlantic meridional overturning circulation, potentially in response to iceberg-derived freshwater injections in the North Atlantic (Kageyama et al., 2010). A few paleoclimatic studies (Dokken and Jansen, 1999; Rasmussen and Thomsen, 2004; Dokken et al., 2013) and sensitivity tests performed with atmospheric models (Li et al., 2010) have also suggested that the expansion of sea ice in the Nordic Seas during GS could be a key amplifier, explaining the large 5-16 °C magnitude of Greenland cooling (Kindler et al., 2014). However, cold sea-surface temperatures (SST) and expansion of sea ice during GS were mainly inferred from indirect marine proxy records, such as significant increases in ice-rafted debris concentration or variations in the relative abundance and oxygen isotopic content of the polar planktonic foraminifera *Neogloboquadrina pachyderma* sinistral coiling (NPS) (Bond and Lotti, 1995; Dokken and Jansen, 1999; Rasmussen and Thomsen, 2004; Dokken et al., 2013) whose preferential depth habitat lies from a few tens of meters to around 250 meters water depth in the Nordic Seas (e.g. Simstich et al., 2003). The occurrence of a pycnocline separating this cold and sea ice-covered surface layer from warmer Atlantic subsurface waters have also been reported during GS on the basis of these and other planktonic foraminifera data supported by benthic foraminifera ones however sometimes interpreted in different ways (e.g. Rasmussen and Thomsen, 2004; Dokken et al., 2013). In parallel, the few direct but qualitative sea-ice reconstructions based on lipid biomarker analyses (Müller and Stein, 2014; Hoff et al., 2016) yielded contrasting interpretations. Here, we provide direct and quantitative reconstructions of variations of sea-surface conditions from a compilation of three Norwegian Sea cores and one northern Northeast Atlantic core strategically positioned across the Faeroe-Iceland Ridge to track rapid hydrographic changes (Dokken and Jansen, 1999; Eynaud et al., 2002; Rasmussen and Thomsen, 2004; Dokken et al., 2013) (Fig. 1A and Table S1). We focus on Marine Isotopic Stage 3 (MIS 3, 30-48 ka cal BP), when millennial variability is strongly imprinted, and accurate chronologies can be established (Austin and Hibbert, 2012). In parallel to these reconstructions, we also use subsurface paleohydrographical data, freshwater hosing simulations and ice core-derived atmospheric data to assess the ocean-cryosphere-atmosphere interactions associated with this abrupt climate variability.

**2        Methods**

**2.1        Stratigraphy**

For the four studied cores, new age models have been established on the basis of radiocarbon AMS $^{14}$C dates coupled to additional tie-points obtained by correlating their magnetic susceptibility records with the NGRIP $\delta^{18}$O signal (North Greenland Ice Core Project members, 2004) (GICC05 time scale; Svensson et al., 2008). This approach is in line with the current consensus that, in this region, increases (respectively decreases) in magnetite content (here, magnetic susceptibility reflecting deep sea currents strength; Kissel et al., 1999) are synchronous with the onset of GI (respectively onset of GS; Kissel et al., 1999; Austin and Hibbert, 2012). Cores MD95-2009, MD95-2010 and MD99-2281 also benefit from additional climate-independent age control points supporting these new age models. A more detailed discussion on the age models can be found in the Supporting Information (Section S1, Fig. S1, and Table S2; Martinson et al., 1987 ; Manthé, 1998; Laj et al., 2004 ; Rasmussen et al., 2006 ; Zumaque et al., 2012 ; Caulle et al., 2013; Reimer et al., 2013; Wolff et al., 2010; Wary et al., 2016).

**2.2        Sea-surface conditions**

Sea-surface conditions are estimated from a transfer function *sensu lato* applied to dinocyst – or dinoflagellate cyst – assemblages using the modern analogue technique (de Vernal and Rochon, 2011) (see Supporting Information Section S2 for further details on the methodology; Rochon et al., 1999; Head et al., 2001; Telford & Birks 2005, 2009, 2011; Telford 2006; Guiot and de Vernal, 2007, 2011a, 2011b; Birks et al., 2010; Radi et al., 2013; de Vernal et al., 2013a,b; Trachsel and Telford, 2016). As dinoflagellates are mostly restricted to the uppermost 50 meters water depth (Sarjeant, 1974), they are assumed to directly reflect sea-surface conditions (see Supporting Information Section S6 for further details). We provide here new sea-surface reconstructions for cores MD95-2009, MD95-2010 and MD99-2281 based on previously published dinocyst counts (Eynaud et al., 2002; Eynaud, 2003a,b; Zumaque et al., 2011) and extend the previously published reconstructions for core MD99-2285 (Wary et al., 2016; see also Wary et al., *in press* for the complete raw dinocyst counts of core MD99-2285). Our statistical approach provides direct and quantitative reconstructions for mean summer and mean winter SST (with, in the present case, root mean square errors of prediction – RMSEP – of 1.5 °C and 1.05 °C respectively), mean summer and mean winter sea-surface salinities (SSS; respective RMSEP of 2.4 and 2.3 psu), and mean annual sea-ice cover (SIC) duration (RMSEP of 1.2 month/year).

**2.3    Model simulations**

We compare our reconstructions with freshwater hosing experiments conducted using five state-of-the-art climate models (Swingedouw et al., 2013). Four of them are coupled ocean-atmosphere models (HadCM3, IPSLCM5A, MPI-ESM, EC-Earth) and one is ocean-only model (ORCA05) (see Supporting Information Section S3 and Table S3; Gordon et al., 2000; Biastoch et al., 2008; Sterl et al., 2012; Dufresne et al., 2013; Jungclaus et al., 2013). One of the models (BCM2) reported in the original study (Swingedouw et al., 2013) has been considered as an outlier and consequently excluded from the present study (see Supporting Information Section S3 for further details). Two types of simulations are considered: (i) the transient control simulations, corresponding to historical simulations without any additional freshwater input, and (ii) the hosing simulations, corresponding to historical simulations with an additional freshwater input of 0.1 Sv released on all the coastal grid points around Greenland with a homogenous rate during 40 years (over the historical era 1965–2004, except for HadCM3 and MPI-ESM for which the experiments were performed over the periods 1960–1999 and 1880–1919, respectively). Several variables have been analyzed: oceanic temperatures (Fig. 1B and 1D), surface (2 m) atmospheric temperatures (Fig. 1C), and barotropic stream function (Fig. S6). Anomalies were calculated as the differences between hosing and control experiments averaged over the 4th decade.

Earlier studies have shown that the response (spatial pattern, amplitudes, …) to freshwater discharges in the North Atlantic depends on several factors including climatic boundary conditions (Swingedouw et al., 2009; Kageyama et al., 2010). Differences of sensitivity to freshwater perturbations in Last Glacial Maximum (LGM) conditions compared to interglacial conditions have been mainly ascribed to differences in ice-sheet and sea-ice configurations. As millennial climatic variability is strongest during MIS 3, it would have been optimal to compare our MIS 3 data to simulations run under MIS 3 conditions rather than pre-industrial ones. However, MIS 3 boundary conditions, and especially cryospheric conditions, are poorly constrained and set at an intermediate level between LGM and present-day boundary conditions (Van Meerbeeck et al., 2009). Nevertheless, it will be worth comparing our reconstructions with MIS 3 simulations conducted using the same state-of-the-art multi-model approach with standardized volume and duration of freshwater flux as soon as such simulations will be available.

**2.4    Complementary data**

To complement our view of the system, we also compare our sea-surface hydrographical reconstructions with (i) the relative abundance of the mesopelagic polar planktonic foraminifera NPS obtained in the same cores (Eynaud et al., 2002; Zumaque et al., 2012; Wary, 2015) and considered as tracer of cold subsurface conditions (see Supporting Information Sections S5 and S6 for further details; Carstens and Wefer, 1992; Bauch et al., 1997; Carstens et al., 1997; Hillaire-Marcel and Bilodeau, 2000; Volkmann and Mensch, 2001; Simstich et al., 2003; Hillaire-Marcel et al., 2004; Kretschmer et al., 2016), and (ii) Greenland ice core deuterium excess data as indicator of Greenland moisture origin (Masson-Delmotte et al., 2005).

**3        Results and Discussion**

Our sea-surface reconstructions reveal contrasted responses of the southeastern Nordic Seas compared to the northeastern Atlantic (Fig. 2, Tables 1, 2 and 3). The Atlantic core MD99-2281 exhibits lower SST during GS compared to GI, and a very short SIC duration throughout MIS3. Paradoxically, the three Norwegian Sea cores record higher SST and shorter SIC durations during the cold North Atlantic GS, and lower SST and longer SIC durations during the warm North Atlantic GI. This atypical pattern is robustly observed in all the three Norwegian Sea sequences, despite distinct physiographical contexts, and strongly expressed in the 63°N cores. At this latitude, SST is systematically anti-correlated against Greenland and North Atlantic temperatures (Table 3), and shows large positive mean annual anomalies in GS compared to GI from +1.7 °C (MD95-2009) to +3.7 °C (MD99-2285) (see Supporting Information Section S4 for details on the calculation of anomalies; Wolff et al., 2010). Despite lower resolution and sensitivity, SST records from MD95-2010 also denote a positive GS mean annual SST anomaly (+0.9 °C), and cooling during GI is further supported by increases in the relative percentage of the polar, sea-ice linked dinocyst *Islandinium minutum* (% I.MIN; Supporting Information Section S2 and Figs. S2 and S3; Rochon et al., 1999; Radi et al., 2013; Heikkilä et al., 2014, 2016). Previous paleoclimatic studies (e.g. de Vernal et al., 2006) evidenced a similar regional SST seesaw pattern during the LGM, with also sometimes warmer than modern SST in the Nordic Seas, suggesting that such a situation might represent a regular feature for glacial periods.

In order to investigate the mechanisms involved in this regional seesaw, we analyze the multi-model freshwater hosing simulations from Swingedouw et al. (2013). The five-member ensemble mean of the differences between hosing and control experiments shows large surface warming in the Nordic Seas while the rest of the North Atlantic surface is strongly cooled in response to freshwater input around Greenland (Fig. 1B). This regional seesaw pattern is robust in the five individual simulations and consistent with concomitant atmospheric cooling above Greenland (Fig. 1C). While the simulated multi-model mean surface warming is weaker than the paleodata-derived one, some individual simulations produce SST increase of up to 4.2 °C in the Nordic Seas (Swingedouw et al., 2013). The multi-model simulations also depict significant sea-ice retreat in the Nordic Seas and sea-ice expansion in the Atlantic sector and Labrador Sea (see Fig. 10 in Swingedouw et al., 2013).

An earlier modelling study (Kleinen et al., 2009) also depicted surface warming of the Nordic Seas in response to a freshwater perturbation, independently from the location of the freshwater input. It was attributed to the subsurface propagation of warm Atlantic water masses beneath the cold North Atlantic meltwater lid (resulting from the freshwater input) up to the Nordic Seas where they re-emerge and mix with ambient waters. Our model simulations indeed show a positive subsurface heat anomaly south of the Greenland-Scotland sill, located below the North Atlantic freshwater lid (Fig. 1D). This freshwater lid has two important consequences: (i) it prevents oceanic vertical mixing which normally transfers winter surface cooling (due to atmospheric heat fluxes) into subsurface, and (ii) it induces hydrographical reorganizations where subpolar gyre transport decreases but water-mass transport from the subtropics into the Nordic Seas increases, especially along the eastern North Atlantic boundary (see Hátún et al., 2005, Kleinen et al., 2009 and Fig. S6).

Although simulated here under present-day background conditions, this physical process may have occurred during stadials in response to meltwater release and provides an explanation for the regional seesaw SST and SIC pattern. A few earlier paleoclimate studies have indeed suggested enhanced advection of warm Atlantic waters through the Continental Slope Current (flowing poleward along the eastern North Atlantic boundary)

during stadial intervals (Peck et al., 2008, based on a core from the Porcupine Seabight) in response to a meltwater release detected at GI-GS transitions (see Wary et al., 2016). Compared to the modern climate system, the potentially reduced northward baroclinic volume transport of Atlantic waters associated with a weaker stadial deep-convection in the Nordic Seas could have been counteracted by (i) an increased northward barotropic transport (with compensation through a larger export at the Denmark Strait for instance), (ii) a larger heat transport due to higher temperature anomalies in the source area, and/or (iii) a greater impact of this northward heat transport on Nordic Seas SST thanks to a larger insolation forcing during MIS3 (Berger and Loutre, 1991).

We now consider subsurface information from our records to complement this mechanism (Fig. 2). Consistent with earlier paleoceanographic studies within the Nordic Seas (Rasmussen and Thomsen, 2004) and the North Atlantic (Bond and Lotti, 1995; Rasmussen and Thomsen, 2004; Eynaud et al., 2009; Jonkers et al., 2010), all our cores reveal the occurrence of colder planktonic foraminiferal assemblages during GS, characterized here by nearly 100% of the mesopelagic taxon NPS. This testifies to the presence of cold polar waters (Eynaud et al., 2009) below a few tens of meters of water depth.

Altogether, this implies the following oceanographic situation during GS: a reduced Atlantic meridional overturning circulation due to large meltwater fluxes (related to and/or sustained by iceberg releases), a southward migration of polar waters, a colder and fresher North Atlantic surface, and a small northward subsurface flow of warm Atlantic waters, propagating below the North Atlantic meltwater lid (and below NPS depth habitat) before remerging at the surface of the Norwegian Sea, above colder polar waters (Fig. 3).

During GS, the upper part of the water column (topmost tens of meters) consists of a layer characterized by fairly high temperatures, notably during summer (Table 1), due to increased heat transport associated with

Atlantic waters without heat loss. Dinocyst-derived sea-surface salinities (Table S4) depict relatively low values, around 31.7 psu over the entire study area, which are likely unfavorable to the development of subpolar surface to mid-surface dweller planktonic foraminifera despite fairly high SST (see Section S5 for further details; Tolderlund and Bé, 1971). 
[revised manuscript text omitted]

Birks, H. J. B., Heiri, O., Seppä, H. and Bjune, A. E.: Strengths and weaknesses of quantitative climate reconstructions based on Late-Quaternary biological proxies, Open Ecol. J., 3, 68–110, doi:10.2174/1874213001003020068, 2010.

Bond, G. C. and Lotti, R.: Iceberg discharges into the North Atlantic on millennial time scales during the last glaciation, Science, 267(5200), 1005–1010, 1995.

Bonne, J.-L., Steen-Larsen, H. C., Risi, C., Werner, M., Sodemann, H., Lacour, J.-L., Fettweis, X., Cesana, G., Delmotte, M., Cattani, O., Vallelonga, P., Kjær, H. A., Clerbaux, C., Sveinbjörnsdõttir, A. E. and Masson-Delmotte, V.: The summer 2012 Greenland heat wave: In situ and remote sensing observations of water vapor 275 isotopic composition during an atmospheric river event, J. Geophys. Res. Atmospheres, 120(7), 2970–2989, doi:10.1002/2014JD022602, 2015.

Carstens, J. and Wefer, G.: Recent distribution of planktonic foraminifera in the Nansen Basin, Arctic Ocean, Deep Sea Res. Part Oceanogr. Res. Pap., 39(2 PART 1), S507–S524, doi:10.1016/S0198-0149(06)80018-X, 1992.

Carstens, J., Hebbeln, D. and Wefer, G.: Distribution of planktic foraminifera at the ice margin in the Arctic (Fram Strait), Mar. Micropaleontol., 29(3–4), 257–269, 1997.

Caulle, C., Penaud, A., Eynaud, F., Zaragosi, S., Roche, D. M., Michel, E., Boulay, S. and Richter, T.: Sea-surface hydrographical conditions off South Faeroes and within the North-Eastern North Atlantic through MIS 2: The response of dinocysts, J. Quat. Sci., 28(3), 217–228, doi:10.1002/jqs.2601, 2013.

de Vernal, A. and Rochon, A.: Dinocysts as tracers of sea-surface conditions and sea-ice cover in polar and subpolar environments, vol. 14., 2011.

de Vernal, A., Rosell-Melé, A., Kucera, M., Hillaire-Marcel, C., Eynaud, F., Weinelt, M., Dokken, T., Kageyama, M.: Comparing proxies for the reconstruction of LGM sea-surface conditions in the northern North Atlantic. Quaternary Science Reviews 25, 2820–2834, 2006.

de Vernal, A., Hillaire-Marcel, C., Rochon, A., Fréchette, B., Henry, M., Solignac, S., Bonnet, S.: Dinocyst-based reconstructions of sea ice cover concentration during the Holocene in the Arctic Ocean, the northern North Atlantic Ocean and its adjacent seas. Quaternary Science Reviews 79, 111–121. doi:10.1016/j.quascirev.2013.07.006, 2013a.

de Vernal, A., Rochon, A., Fréchette, B., Henry, M., Radi, T., Solignac, S.: Reconstructing past sea ice cover of
the Northern Hemisphere from dinocyst assemblages: Status of the approach. Quaternary Science Reviews 79, 122–134, 2013b.

Dokken, T. M. and Jansen, E.: Rapid changes in the mechanism of ocean convection during the last glacial period, Nature, 401(6752), 458–461, 1999.

Dokken, T. M., Nisancioglu, K. H., Li, C., Battisti, D. S. and Kissel, C.: Dansgaard-Oeschger cycles:
Interactions between ocean and sea ice intrinsic to the Nordic seas, Paleoceanography, 28(3), 491–502, 2013.

Dufresne, J. L., Foujols, M. A., Denvil, S., Caubel, A., Marti, O., Aumont, O., Balkanski, Y., Bekki, S., Bellenger, H., Benshila, R., Bony, S., Bopp, L., Braconnot, P., Brockmann, P., Cadule, P., Cheruy, F., Codron, F., Cozic, A., Cugnet, D., de Noblet, N., Duvel, J. P., Ethé, C., Fairhead, L., Fichefet, T., Flavoni, S., Friedlingstein, P., Grandpeix, J. Y., Guez, L., Guilyardi, E., Hauglustaine, D., Hourdin, F., Idelkadi, A., Ghattas,
J., Joussaume, S., Kageyama, M., Krinner, G., Labetoulle, S., Lahellec, A., Lefebvre, M. P., Lefevre, F., Levy, C., Li, Z. X., Lloyd, J., Lott, F., Madec, G., Mancip, M., Marchand, M., Masson, S., Meurdesoif, Y., Mignot, J., Musat, I., Parouty, S., Polcher, J., Rio, C., Schulz, M., Swingedouw, D., Szopa, S., Talandier, C., Terray, P., Viovy, N. and Vuichard, N.: Climate change projections using the IPSL-CM5 Earth System Model: from CMIP3 to CMIP5, Clim. Dyn., 40(9–10), 2123–2165, doi:10.1007/s00382-012-1636-1, 2013.

Ehlers, J. and Gibbard, P. L.: The extent and chronology of Cenozoic Global Glaciation, Quat. Int., 164–165, 6–20, 2007.

Elliot, M., Labeyrie, L., Dokken, T. and Manthe, S.: Coherent patterns of ice-rafted debris deposits in the Nordic regions during the last glacial (10-60 ka), Earth Planet. Sci. Lett., 194(1–2), 151–163, doi:10.1016/S0012-821X(01)00561-1, 2001.

Eynaud, F.: Dinoflagellate cysts counts of sediment core MD95-2009, PANGAEA, Unpublished dataset #94393, 2003a.

Eynaud, F.: Dinoflagellate cysts counts of sediment core MD95-2010, PANGAEA, doi:10.1594/PANGAEA.94394, 2003b.

Eynaud, F., Turon, J. L., Matthiessen, J., Kissel, C., Peypouquet, J. P., De Vernal, A. and Henry, M.: Norwegian
sea-surface palaeoenvironments of marine oxygen-isotope stage 3: The paradoxical response of dinoflagellate cysts, J. Quat. Sci., 17(4), 349–359, doi:10.1002/jqs.676, 2002.

[revised manuscript text omitted]

Müller, J., Stein, R.: High-resolution record of late glacial and deglacial sea ice changes in Fram Strait corroborates ice–ocean interactions during abrupt climate shifts. Earth and Planetary Science Letters 403, 446–455, 2014.

North Greenland Ice Core Project members: High-resolution record of Northern Hemisphere climate extending into the last interglacial period, Nature, 431(7005), 147–151, doi:10.1038/nature02805, 2004.

Peck, V.L., Hall, I.R., Zahn, R., Elderfield, H.: Millennial-scale surface and subsurface paleothermometry from the northeast Atlantic, 55-8 ka BP. Paleoceanography 23. doi:10.1029/2008PA001631, 2008.

Radi, T., Bonnet, S., Cormier, M. A., de Vernal, A., Durantou, L., Faubert, É., Head, M. J., Henry, M., Pospelova, V., Rochon, A. and Van Nieuwenhove, N.: Operational taxonomy and (paleo-)autecology of round, brown, spiny dinoflagellate cysts from the Quaternary of high northern latitudes, Mar. Micropaleontol., 98, 41–57, 2013.

Rasmussen, S. O., Andersen, K. K., Svensson, A. M., Steffensen, J. P., Vinther, B. M., Clausen, H. B., Siggaard-Andersen, M.-L., Johnsen, S. J., Larsen, L. B., Dahl-Jensen, D., Bigler, M., Röthlisberger, R., Fischer, H., Goto-Azuma, K., Hansson, M. E. and Ruth, U.: A new Greenland ice core chronology for the last glacial termination, J. Geophys. Res., 111(D6), doi:10.1029/2005JD006079, 2006.

Rasmussen, T. L. and Thomsen, E.: The role of the North Atlantic Drift in the millennial timescale glacial climate fluctuations, Palaeogeogr. Palaeoclimatol. Palaeoecol., 210(1), 101–116, doi:10.1016/j.palaeo.2004.04.005, 2004.

Reimer, P. J., Bard, E., Bayliss, A., Beck, J. W., Blackwell, P. G., Bronk Ramsey, C., Buck, C. E., Cheng, H., Edwards, R. L. and Friedrich, M.: IntCal13 and Marine13 radiocarbon age calibration curves 0-50,000 years cal BP, Radiocarbon, 55(4), 1869–1887, 2013.

Rochon, A., de Vernal, A., Turon, J.-L., Matthiessen, J. and Head, M. J., Eds.: Distribution of dinoflagellate cysts in surface sediments from the North Atlantic Ocean and adjacent basins and quantitative reconstruction of sea-surface parameters, AASP special pub., Dallas, Texas., 1999.

Sarjeant, W. A. S.: Fossil and living dinoflagellates, Academic Press Inc., London, UK., 1974.

Simstich, J., Sarnthein, M. and Erlenkeuser, H.: Paired δ18O signals of Neogloboquadrina pachyderma (s) and Turborotalita quinqueloba show thermal stratification structure in Nordic Seas, Mar. Micropaleontol., 48(1–2), 107–125, 2003.

Sterl, A., Bintanja, R., Brodeau, L., Gleeson, E., Koenigk, T., Schmith, T., Semmler, T., Severijns, C., Wyser, K. and Yang, S.: A look at the ocean in the EC-Earth climate model, Clim. Dyn., 39(11), 2631–2657, doi:10.1007/s00382-011-1239-2, 2012.

Svensson, A., Andersen, K. K., Bigler, M., Clausen, H. B., Dahl-Jensen, D., Davies, S. M., Johnsen, S. J., Muscheler, R., Parrenin, F., Rasmussen, S. O., Röthlisberger, R., Seierstad, I., Steffensen, J. P. and Vinther, B. M.: A 60 000 year Greenland stratigraphic ice core chronology, Clim. Past, 4(1), 47–57, doi:10.1126/science.305.5690.1567a, 2008.

Swingedouw, D., Mignot, J., Braconnot, P., Mosquet, E., Kageyama, M. and Alkama, R.: Impact of freshwater release in the North Atlantic under different climate conditions in an OAGCM, J. Clim., 22(23), 6377–6403, 2009.

Swingedouw, D., Rodehacke, C. B., Behrens, E., Menary, M., Olsen, S. M., Gao, Y., Mikolajewicz, U., Mignot, J. and Biastoch, A.: Decadal fingerprints of freshwater discharge around Greenland in a multi-model ensemble, Clim. Dyn., 41(3–4), 695–720, doi:10.1007/s00382-012-1479-9, 2013.

Tolderlund, D. S. and Bé, A. W. H.: Seasonal Distribution of Planktonic Foraminifera in the Western North Atlantic, Micropaleontology, 17(3), 297–329, doi:10.2307/1485143, 1971.

Telford, R.J.: Limitations of dinoflagellate cyst transfer functions, Quaternary Sci. Rev., 25, 1375–1382, doi:10.1016/j.quascirev.2006.02.012, 2006.

Telford, R. J. and Birks, H. J. B.: The secret assumption of transfer functions: problems with spatial autocorrelation in evaluating model performance, Quaternary Sci. Rev., 24, 2173–2179, doi:10.1016/j.quascirev.2005.05.001, 2005.

Telford, R. J. and Birks, H. J. B.: Evaluation of transfer functions in spatially structured environments, Quaternary Sci. Rev., 28, 1309–1316, doi:10.1016/j.quascirev.2008.12.020, 2009.

Telford, R. J. and Birks, H. J. B.: QSR Correspondence "Is spatial autocorrelation introducing biases in the apparent accuracy of palaeoclimatic reconstructions?", Quaternary Sci. Rev., 30, 3210–3213, doi:j.quascirev.2011.07.019, 2011.

Trachsel, M. and Telford, R. J.: Technical note: Estimating unbiased transfer-function performances in spatially structured environments, Clim. Past, 12, 1215-1223, doi:10.5194/cp-12-1215-2016, 2016.

Van Meerbeeck, C. J., Renssen, H. and Roche, D. M.: How did marine isotope stage 3 and Last Glacial Maximum climates differ? Perspectives from equilibrium simulations, Clim. Past, 5(1), 33–51, 2009.

de Vernal, A., Rosell-Melé, A., Kucera, M., Hillaire-Marcel, C., Eynaud, F., Weinelt, M., Dokken, T. and 450 Kageyama, M.: Comparing proxies for the reconstruction of LGM sea-surface conditions in the northern North Atlantic, Quat. Sci. Rev., 25(21–22), 2820–2834, 2006.

Volkmann, R. and Mensch, M.: Stable isotope composition ($\delta18O$, $\delta13C$) of living planktic foraminifers in the outer Laptev Sea and the Fram Strait, Mar. Micropaleontol., 42(3–4), 163–188, doi:10.1016/S0377-8398(01)00018-4, 2001.

Wary, M.: Rôle des conditions océaniques et des ice-shelves en périphérie des calottes européennes lors des évènements climatiques abrupts de la dernière période glaciaire, PhD thesis, Université de Bordeaux. [online] Available from: www.theses.fr/2015BORD0316/document, 2015.

Wary, M., Eynaud, F., Rossignol, L., Lapuyade, J., Gasparotto, M.-C., Londeix, L., Malaizé, B., Castéra, M.-H. and Charlier, K.: Norwegian Sea warm pulses during Dansgaard-Oeschger stadials: Zooming in on these 460 anomalies over the 35–41 ka cal BP interval and their impacts on proximal European ice-sheet dynamics, Quat. Sci. Rev., 151, 255–272, doi:http://dx.doi.org/10.1016/j.quascirev.2016.09.011, 2016.

Wary, M., Eynaud, F., Zaragosi, S., Rossignol, L., Sabine, M., Castéra, M.-H., Billy, I.: The Southern Norwegian Sea during the last 45 ka: hydrographical reorganizations under changing ice-sheet dynamics. Journal of Quaternary Science, in press.

Wolff, E. W., Chappellaz, J., Blunier, T., Rasmussen, S. O. and Svensson, A.: Millennial-scale variability during the last glacial: The ice core record, Quat. Sci. Rev., 29(21–22), 2828–2838, 2010.

Zumaque, J., Eynaud, F., Zaragosi, S., Marret, F., Matsuzaki, K. M., Kissel, C., Roche, D. M., Malaizé, B., Michel, E., Billy, I., Richter, T. and Palis, E.: An ocean–ice coupled response during the last glacial: a view from a marine isotopic stage 3 record south of the Faeroe Shetland Gateway, Clim. Past, 8(6), 1997–2017, 2012.

[Figure]

**Figure 1: Hydrographical context and five-member ensemble mean of temperature anomalies between hosing and control experiments. (a) Schematic surface current pattern (STG, subtropical gyre; SPG, subpolar gyre; CSC, Continental Slope Current; NAC, North Atlantic Current; EIC, East Icelandic Current). (b, c) Five-member ensemble mean of SST (b) and surface atmospheric temperature (c) anomalies (°C). (d) Latitude-depth section of the five-member ensemble mean of oceanic temperature anomalies (°C, zonal average over Atlantic ocean). Also shown are the locations of the studied marine cores (MD95-2010, MD95-2009, MD99-2285, MD99-2281) and Greenland ice cores (NGRIP, GRIP). Black dashes indicate grid points where all models converge on the anomaly sign.**

[Figure]

**Figure 2: Proxy records. (a) %NPS (shading in MD99-2281 at an arbitrary threshold to better illustrate changes). (b to e) SST and SIC records, shaded relatively to the mean value (indicated in gray) of the parameter over the studied period. Error bars are shown in panel (c). %I.MIN of MD95-2010 is also shown. (f) GRIP deuterium excess record and associated reconstructed source temperature anomaly (compared to modern value) of the evaporative source region for Greenland precipitation, assuming no change in humidity (Masson-Delmotte et al., 2005) (shaded relatively to its mean value over the studied period). (g) NGRIP δ¹⁸O (GICC05 age scale; North Greenland Ice Core Project members, 2004; Svensson et al., 2008). Gray bands highlight stadial periods.**

[Figure]

**Figure 3: Conceptual hydrographical scheme. The diagrams depict the mean conditions in the subboreal Atlantic during stadials (left) /interstadials (right), and summer (middle panels) /winter (lower panels). Section location is indicated on the top maps. Bathymetry is from GEBCO (www.gebco.net), and has been simplified for sections. Ice-sheet extent on maps corresponds to the Last Glacial Maximum extension (Ehlers and Gibbard, 2007). Colors indicate temperature range, as indicated by the bottom scale. Potential depth range (Simstich et al., 2003) and optimal temperature range (Tolderlund and Bé, 1971) of NPS habitat, whose main production period occurs in summer in the Nordic Seas (Simstich et al., 2003), are also indicated.**

**Table 1: SST anomalies.**

| Core | Number of samples | | GS SST (°C) | | | GI SST (°C) | | | Mean annual SST anomalies (GS-GI; °C) |
|------|----|----|-------------|-------------|-------------|-------------|-------------|-------------|----|
| | GS | GI | mean winter | mean summer | mean annual | mean winter | mean summer | mean annual | |
| MD99-2281 | 23 | 39 | 0.9 | 14.6 | 7.8 | 1.5 | 14.4 | 8.0 | -0.2 |
| MD99-2285 | 26 | 22 | 0.9 | 10.9 | 5.9 | -0.6 | 4.9 | 2.2 | 3.7 |
| MD95-2009 | 12 | 17 | 0.3 | 11.0 | 5.6 | -0.4 | 8.3 | 4.0 | 1.7 |
| MD95-2010 | 6 | 9 | 0.6 | 13.4 | 7.0 | 0.2 | 12.4 | 6.1 | 0.9 |

**Table 2: SIC duration anomalies.**

| Core | Number of samples | | GS SIC (mth/yr) mean annual | GI SIC (mth/yr) mean annual | Mean annual SIC anomalies (GS-GI; mth/yr) |
|---|---|---|---|---|---|
| | **GS** | **GIS** | | | |
| MD99-2281 | 23 | 39 | 0.9 | 0.6 | 0.3 |
| MD99-2285 | 26 | 22 | 3.2 | 6.2 | -3.0 |
| MD95-2009 | 12 | 17 | 3.4 | 4.4 | -1.0 |
| MD95-2010 | 6 | 9 | 2.0 | 2.7 | -0.7 |

**Table 3: Correlation coefficients over the 48-30 ka cal BP interval between Greenland temperatures (NGRIP δ$^{18}$O; North Greenland Ice Core Project members, 2004; Svensson et al., 2008), North Atlantic (MD99-2281) and Norwegian Sea (MD99-2285, MD95-2009, MD95-2010) winter SST.**

| | | winter SST | | | |
| | | MD99-2281 | MD99-2285 | MD95-2009 | MD95-2010 |
|---|---|---|---|---|---|
| | NGRIP δ$^{18}$O | 0,24 | -0,45 | -0,42 | -0,10 |
| | MD99-2281 | | -0,31 | -0,31 | 0,18 |
| winter SST | MD99-2285 | | | 0,59 | -0,08 |
| | MD95-2009 | | | | -0,11 |

**S1. Stratigraphy**

For the present study, new age models have been developed using the same state-of-the-art approach for each of the studied cores. Each age model has been elaborated by combination of two types of control points: (i) radiocarbon (AMS $^{14}$C) datings (Manthé, 1998; Dokken and Jansen, 1999; Zumaque et al., 2012; Caulle et al., 2013; Wary et al., 2016) converted to calendar ages using Calib 7.1.0 calibration program (http://calib.qub.ac.uk/calib/) and Marine13 calibration curve (Reimer et al., 2013), with an integrated 405 year marine reservoir correction (Table S2), and (ii) event-based tie-points derived from the correlation of the magnetic susceptibility signals to the NGRIP-GICC05 $\delta^{18}$O record (North Greenland Ice Core Project members, 2004; Svensson et al., 2008) (i.e. the recommended North Atlantic regional stratotype after Austin and Hibbert, 2012). The rationale is that marine records of magnetic parameters from MIS 3 are consistent across the North Atlantic basin along the path of different overflow branches of the North Atlantic Deep Water and can be tied to the high frequency climatic variability (Dansgaard-Oeschger – DO – cycles) characteristic of this period (Kissel et al., 1999). Chronostratigraphies of the cores have been constructed using this DO event chronostratigraphy after dates from Wolff et al. (2010) (NGRIP-GICC05 derived, see Table S2 and Fig. S1). Core MD95-2010 also benefits from the recovery of ash-layer well-dated horizons (Dokken and Jansen, 1999) (http://doi.pangaea.de/10.1594/PANGAEA.735730; Table S2) used as additional control points. Each age model has been finally established on the basis of a linear interpolation between ages and tie-points.

Stratigraphies of cores MD95-2009 and MD99-2281 are additionally supported by the occurrence of supplementary tie-points (not used but fitting our age model constructions), independent from climate, derived from the record of the changes in the Earth's magnetic field intensity, namely the two prominent lows attributed to the Mono Lake and the

Laschamp excursions (~34.5 and ~41 ka cal BP respectively; see Kissel et al., 1999; Laj et al., 2004; Zumaque et al., 2012).

It is worth mentioning that we used the Calib 7.1.0 integrated reservoir age to correct

AMS $^{14}$C datings rather than using more accurate reservoir age estimates because such high temporal data are not available at the moment for the whole 40-10 ka BP interval in the Norwegian Sea, and even if such data were available, using them would not change our results nor interpretations since, on the time interval discussed, age models are primarily constrained by event-based tie-points and supported by an additional stratigraphic control independent from climate.

**S2. Dinocyst counts, transfer function and seasonality signals**

Dinocysts were counted in the 10-150 µm fraction after classical palynological preparation         of         sediment         samples         (http://www.epoc.u- bordeaux.fr/index.php?lang=fr&page=eq_paleo_pollens). When possible, a minimum of

300 dinocysts were counted in each sample using a Leica Microscope at x400

magnification. Species identification follows (Rochon et al., 1999; Head et al., 2001;

Radi et al., 2013). Relative abundances of each species were calculated relative to the total sum of Quaternary dinocysts. Among the dominant species of the four studied cores, one deserves here a special attention – *Islandinium minutum* – as it is strongly related to cold and seasonally sea-ice covered surface environments (Radi et al., 2013) where this heterotrophic taxon can exhibit a complex spatial and temporal dynamic tightly linked to nutrient and prey availability (e.g. Heikkilä et al., 2014, 2016). Its highest abundances are observed in areas covered with sea ice between 8 and 12 months/year (Rochon et al.,

1999) (Fig. S2). In the Norwegian Sea cores, records of %I.MIN (Fig. S3) clearly indicate lower SST and longer SIC during GI, and milder surface conditions during GS. In the

Atlantic core MD99-2281, the %I.MIN signal show low values throughout the studied period, indicative of a strongly reduced SIC duration; the very slight increase of %I.MIN

during GS indicate relatively colder sea-surface conditions, thus confirming the difference of pattern observed in-between the Nordic Seas and the North Atlantic Ocean.

Past sea-surface conditions were derived from a transfer function applied to dinocyst assemblages, using the modern analogue technique (see Guiot and de Vernal, 2007,

2011a, 2011b for a review of this technique). Briefly, calculation relies on a statistical comparison of fossil samples to a large set of modern (surface sediment) samples. The five best analogues (i.e. minimal statistical dissimilarity between the species spectra) are selected for the reconstructions. The hydrographic data corresponding to these analogues, compiled from the 2001 version of the World Ocean Atlas for SST and sea-surface salinities (extracted at 10 meters water depth) and from the National Snow and Ice Data

Center (NSIDC) of Boulder for sea ice data, are then used to calculate weighted (inversely to the dissimilarity of the analogues) averaged past sea-surface parameters.

Quantitative reconstructions, showing similar patterns than those discussed here, were previously published for our studied cores: for cores MD95-2009 and MD95-2010 using a modern dinocyst database including 677 samples (Eynaud et al., 2002), for core MD99-

2281 using the 1189 modern sample database (Zumaque et al., 2012), and for the 35-41

ka cal BP interval of core MD99-2285 using the extended 1207 modern sample database (Wary et al., 2016).

For the present study, the published dinocyst counts of those cores and the new data of core MD99-2285 were statistically treated with this extended modern database composed of 1207 sites from North Atlantic Ocean, Arctic and sub-Arctic basins, Mediterranean

Sea and North Pacific Ocean (database available from the DINO9 workshop, http://pcwww.liv.ac.uk/~dino9/workshops.htm), to provide quantitative reconstructions for hydrological parameters. These include mean summer (July-August-September) and mean winter (January-February-March) SST (respective RMSEP of 1.5 °C and 1.05 °C), mean summer and mean winter SSS (respective RMSEP of 2.4 and 2.3 psu), and mean annual SIC duration (RMSEP of 1.2 month/year). Seasonality signals (Fig. S4) were then determined by subtracting mean winter SST to mean summer SST.

The performance of this statistical treatment has been highly criticized through several successive works (Telford 2006; Telford & Birks 2005, 2009, 2011; Birks et al., 2010;

Trachsel and Telford, 2016) arguing that spatial autocorrelation of the hydrographical parameters in the modern database violates one of the basic assumptions of transfer functions and leads to over-optimistic estimates of the prediction power of this technique.

However, parallel studies equally based on cross-validation schemes (Guiot and de

Vernal, 2011a,b; de Vernal et al., 2013a,b) showed that this spatial autocorrelation has in fact relatively low impact on the calculation of the error of prediction of the MAT

transfer function applied to dinocyst assemblages. To ensure that one can assess by his own the reliability and robustness of our reconstructions, the distance to the nearest analogue (i.e. a reconstruction diagnostic tool) is provided in Fig. S5. Nonetheless, we would like to stress that our interpretations are primarily based on the ecological message brought by our raw dinocyst assemblages (cf. Fig. S2 and S3), and that higher RMSEP

would therefore not affect them.

We chose to graphically represent winter SST (Fig. 2) where GI-GS amplitudes are comparable within the four cores, but note that the same patterns are observable in annual

SST records (i.e. average of the seasonal signals) from all the four cores (Fig. S5). These same patterns are also observable in summer SST records from the three Norwegian Sea cores (with even higher amplitudes), but not from the Atlantic core MD99-2281 where an opposite pattern seems to be recorded but with a significantly smaller amplitude as compared to winter SST record. We attribute this behavior to the nodal location of this site, i.e. in the transitional area where (i) during GS, warm Atlantic subsurface waters re- emerge at the surface, and (ii) during GI, the polar front migrates northward/southward during summer/winter (cf. Fig. 3).

**S3. Model simulations**

Freshwater hosing experiments were conducted using four coupled ocean-atmosphere models (HadCM3, IPSLCM5A, MPI-ESM, EC-Earth) and one ocean-only model (ORCA05; Table S3). Results from the coupled ocean-atmosphere BCM2 model are not considered here, while reported in the original study (Swingedouw et al., 2013). It showed a very different behavior as compared to the five models considered here, and was therefore considered as an outlier in the former ensemble. According to the authors of the original study, this is linked to the fact that the freshwater spread exhibits a different path in BCM2 compared to the five other models, probably in relation with the very low resolution in the ocean in BCM2 (around 3° in the North Atlantic, *cf.* their Fig.

1) compared to the others (from less than 0.5°C to around 1° in the North Atlantic).

Nevertheless, considering a multi-model result including BCM2 only slightly changes the pattern of the response and its amplitude.

Note that results from the ocean-only model (ORCA05) display strong similarities with the ones from the four coupled ocean-atmosphere models. It implies that the structure of the simulated changes is mostly driven by oceanic processes and weakly due to atmospheric feedbacks.

**S4. SST, SIC and SSS anomalies**

SST, SIC duration, and SSS anomalies were calculated for each core (respectively Table

1, Table 2, and Table S4) as follows: mean winter and mean summer SST and SSS, as well mean annual SIC duration, were determined over (i) the stadial periods, comprising

GS 10, 9, 8, 7 and 6, and (ii) the interstadial periods, comprising GI 10, 9, 8, 7, 6, and 5, using the GICC05 age limits of each GS/GI from Wolff et al. (2010). Then, mean annual

SST (and SSS), for GS and for GI periods, were determined by averaging mean winter and mean summer SST (SSS). Finally, mean annual SST (SSS and SIC) anomalies were obtained by subtracting mean annual GI SST (SSS and SIC) to mean annual GS SST

(SSS and SIC; i.e. GS minus GI).

**S5. Relative percentages of *Neogloboquadrina pachyderma* sinistral coiling (%NPS)**

Planktonic foraminifera were counted in the $> 150$ µm fraction after classical preparation of sediment samples (washed through a 150 µm sieve before being dried). When possible, a minimum of 300 specimens were counted in each sample, and relative abundances of each species were determined (Eynaud et al., 2002; Zumaque et al., 2012;

Wary, 2015), revealing nearly continuous monospecific dominance of NPS (except in core MD99-2281), a taxon usually used to track the migration of cold polar waters (Eynaud et al., 2009).

It is worth mentioning that the absence of subpolar surface to mid-surface dweller planktonic foraminifera (e.g. *T. quinqueloba* and *G. bulloides*) in the Norwegian Sea during GS is consistent with our reconstructions and interpretations since, despite high enough SST, SSS were likely too low (see Table S4) for the development of these taxa according to their ecological tolerances (e.g. Tolderlund and Bé, 1971).

**S6. Dinoflagellates versus NPS depth habitats**

Noticeable differences exist in the depth habitats of dinoflagellates and NPS, which, if not taken into account, can lead to inaccurate interpretations. Dinoflagellates are known to be mainly restricted to the uppermost 50 meters of the water column (Sarjeant et al.,

1974), with autotrophic organisms restricted to the photic layer and heterotrophic organisms feeding on autotrophic organisms. This implies that dinocysts are tracers of sea-surface *stricto sensu* conditions. On the contrary, growing evidence of the mesopelagic affinity characterizing the planktonic foraminifera NPS has emerged during the last decades (Carstens and Wefer, 1992; Bauch et al., 1997; Carstens et al., 1997;

Hillaire-Marcel and Bilodeau, 2000; Volkmann and Mensch, 2001; Simstich et al., 2003;

Hillaire-Marcel et al., 2004; Kretschmer et al., 2016). This involves that dinocysts and

NPS may not track the same water mass, i.e. that NPS may not track sea-surface conditions as often admitted but rather subsurface or near-surface conditions. The present study illustrates well this possibility of decoupling, with (i) in the case of a strong stratification of the upper few hundred meters of the water column (GS), dinocyst and

NPS displaying opposite signals, i.e. tracking different water masses (the surface and subsurface, respectively), and (ii) in the case of reduced/zero stratification, dinocyst and

NPS displaying concordant signals, i.e. tracking the same homogeneous upper water mass.

**Figure S1**. Information regarding the age model construction of the studied cores. (a)

Age versus depth plots of the respective chronological constrains used to built the chronology (Table S2). (b) Alignment of magnetic susceptibility (MS) records of the respective studied cores to NGRIP GICC05 $\delta^{18}$O record (North Greenland Ice Core

Project members, 2004; Svensson et al., 2008). Vertical lines illustrate the position of the tie-points derived from peak matching.

[Figure]

[Figure]

**Figure S2.** *Islandinium minutum* distribution and ecology. (a) *Islandinium minutum*

distribution within the modern dinocyst database made of 1207 points. (b) Oceanic temperatures at 10 mbsl (WOA09 data; Locarnini et al., 2010). (c) Sea-ice cover (with concentration greater than 50%) duration within the modern dinocyst database made of

1207 points (after data provided by the National Climate Data Centre in Boulder). These maps demonstrate the strong link of this dinocyst taxon with cold and seasonally sea-ice covered surface environments.

[Figure]

[Figure]

[Figure]

**Figure S3.** Relative percentage of *Islandinium minutum* within the four studied cores. For each core (from a to d) the %I.MIN records, indicative of colder SST and longer SIC, are compared to the magnetic susceptibility signals which can be directly aligned with

Greenland climate variability as detected within (e) NGRIP $\delta^{18}O$ record (North

Greenland Ice Core Project members, 2004; Svensson et al., 2008). Gray bands highlight low NGRIP $\delta^{18}O$ and low MS values, i.e. stadial periods. It is worth noting the opposite scheme described by %I.MIN variations within the Nordic Seas *versus* in the Atlantic sector, which again illustrates the seesaw pattern observed between Nordic Seas and

North Atlantic.

[Figure]

**Figure S4.** Sea-surface seasonality contrasts, calculated in the different cores as summer

SST minus winter SST and plotted along with the reference NGRIP $\delta^{18}O$ stratotype (GICC05 age scale; North Greenland Ice Core Project members, 2004; Svensson et al.,

2008). Sea-surface seasonality values are shaded relatively to the mean value obtained over the studied period within each core (threshold value indicated in gray next to each record). Triangles indicate modern values of summer SST (red), winter SST (blue), and seasonality (black) for each study site (WOA09 data; Locarnini et al., 2010). As in Fig.

S3, gray bands highlight low NGRIP $\delta^{18}O$ and low MS values, i.e. stadial periods.

[Figure]

**Figure S5.** Annual and seasonal SST records and distance to the nearest analogue in the four studied cores plotted along with NGRIP δ¹⁸O record (GICC05 age scale; North

Greenland Ice Core Project members, 2004; Svensson et al., 2008). Triangles indicate modern values of summer and winter SST (WOA09 data; Locarnini et al., 2010). As in

Fig. S3, gray bands highlight low NGRIP δ¹⁸O and low MS values, i.e. stadial periods.

[Figure]

**Figure S6.** Five-member ensemble mean of barotropic stream function anomalies. Colors represent anomalies between hosing and control experiments averaged over the 4[th]

decade. In contours is the control simulation barotropic stream function averaged over the historical era.

[Figure]

**Table S1.** Location of studied marine cores.

| Core | Latitude (°N) | Longitude (°E) | depth (mbsl) |
|---|---|---|---|
| MD95-2010 | 66.68 | 4.57 | 1,226 |
| MD95-2009 | 62.74 | -4.00 | 1,027 |
| MD99-2285 | 62.69 | -3.57 | 885 |
| MD99-2281 | 60.34 | -9.46 | 1,197 |

* mbsl: meters below sea level.

**Table S2.** Age constrains for each of the studied cores.

**Table S3.** Characteristics of the five models considered.

| Model | Institute | Type* | Ocean | Atmosphere | Reference |
|---|---|---|---|---|---|
| HadCM3 | Hadley Centre | OAGCM | No name 1.25 x 1.25, L20 | HadAM3 91 x 76, L19 | (Gordon et al., 2000) |
| IPSLCM5A | Institut Pierre Simon Laplace | OAGCM | NEMO 2°, L31 | LMD5 96 x 96, L39 | (Dufresne et al. 2013) |
| MPI-ESM | MPI | OAGCM (ESM) | MPI-OM 1.5°, L40 | ECHAM6 T63-L47 | (Jungclaus et al., 2013) |
| ORCA05 | GEOMAR | OGCM | NEMO 0.5°, L46 | CORE.v2 Forcing | (Biastoch et al., 2008) |
| EC-Earth | DMI | OAGCM | NEMO 1°, L42 | IFS T159-L31 | (Sterl et al., 2012) |

* OAGCM: Ocean-Atmosphere General Circulation Model; OGCM: Ocean General

Circulation Model; ESM: Earth System Model

**Table S4.** SSS anomalies.

| Core | Number of samples | | GS SSS (psu) | | | GI SSS (psu) | | | Mean annual SSS anomalies (GS-GI; psu) |
|------|------|------|------|------|------|------|------|------|------|
| | GS | GI | mean winter | mean summer | mean annual | mean winter | mean summer | mean annual | |
| MD99-2281 | 23 | 39 | 32.1 | 31.3 | 31.7 | 32.4 | 31.6 | 32.0 | -0.3 |
| MD99-2285 | 26 | 22 | 32.7 | 31.6 | 32.2 | 32.9 | 31.0 | 32.0 | 0.2 |
| MD95-2009 | 12 | 17 | 32.0 | 30.9 | 31.5 | 32.0 | 31.0 | 31.5 | 0.0 |
| MD95-2010 | 6 | 9 | 32.0 | 31.1 | 31.5 | 31.8 | 30.4 | 31.1 | 0.4 |